# GOTHAM: Graph Class Incremental Learning Framework under Weak Supervision

**Aditya Hemant Shahane**                                                          *eez248435@ee.iitd.ac.in*
*Department of Electrical Engineering*
*Indian Institute of Technology Delhi*

**Prathosh A. P.**                                                                *prathosh@iisc.ac.in*
*Department of Electrical Communication Engineering*
*Indian Institute of Science Bengaluru*

**Sandeep Kumar**                                                                 *ksandeep@ee.iitd.ac.in*
*Department of Electrical Engineering*
*Yardi School of Artificial Intelligence*
*Bharti School of Telecommunication Technology and Management*
*Indian Institute of Technology Delhi*

**Reviewed on OpenReview:** *https://openreview.net/forum?id=hCyT4RsF27&noteId=eAOjhejWbB*

## Abstract

Graphs are growing rapidly, along with the number of distinct label categories associated with them. Applications like e-commerce, healthcare, recommendation systems, and various social media platforms are rapidly moving towards graph representation of data due to their ability to capture both structural and attribute information. One crucial task in graph analysis is node classification, where unlabeled nodes are categorized into predefined classes. In practice, novel classes appear incrementally sometimes with just a few labels (seen classes) or even without any labels (unseen classes), either because they are new or haven't been explored much. Traditional methods assume abundant labeled data for training, which isn't always feasible. We investigate a broader objective: *Graph Class Incremental Learning under Weak Supervision (GCL)*, addressing this challenge by meta-training on base classes with limited labeled instances. During the incremental streams, novel classes can have few-shot or zero-shot representation. Our proposed framework GOTHAM efficiently accommodates these unlabeled nodes by finding the closest prototype representation, serving as class representatives in the attribute space. For Text-Attributed Graphs (TAGs), our framework additionally incorporates semantic information to enhance the representation. By employing teacher-student knowledge distillation to mitigate forgetting, GOTHAM achieves promising results across various tasks. Experiments on datasets such as Cora-ML, Amazon, and OBGN-Arxiv showcase the effectiveness of our approach in handling evolving graph data under limited supervision. The code implementation is available here: `https://encr.pw/uYOe2`

## 1 Introduction

Graph-structured data are ubiquitously used in many real-world applications, such as citation graphs (Cummings & Nassar, 2020; Tang et al., 2008), biomedical graphs (Subramanian et al., 2005; Zhai et al., 2023), circuit optimization (Shahane et al., 2023; Hakhamaneshi et al., 2023) and social networks (Qi et al., 2012). Recently, Graph Neural Networks (GNNs) have been proposed (Cao et al., 2016; Subramanian et al., 2005; Henaff et al., 2015; Xu et al., 2019; Kipf & Welling, 2017; Veličković et al., 2018; Hamilton et al., 2017) to

model graph-structured data by leveraging the structural and attributed information along the graph. As a central task in graph machine learning, node classification (Wang et al., 2020b; Xhonneux et al., 2020; Wang et al., 2021b; Zhu et al., 2021) has achieved remarkable progress with the rise of GNNs. While these methods concentrate on static graphs to classify unlabeled nodes into predetermined classes, real-world graphs are dynamic. In practice, graphs grow (You et al., 2022; Lu et al., 2022; Tan et al., 2022) rapidly incorporating nodes and edges belonging to novel classes incrementally. For example, (1) think of a biomedical graph. Each node represents a rare disease category, and edges show how these diseases relate to each other. As new disease categories emerge, they are gradually added to this graph. Such methods significantly aid the drug discovery process. (2) For food delivery systems nodes correspond to various zip codes and the edges indicate the spatial distances between them. As the company expands its reach, new zip codes are systematically incorporated into this graph, facilitating efficient supply chain management. GNNs typically require a large amount of labeled data to learn effective node representations (Ding et al., 2020b; Zhou et al., 2019b). In practice, catching up with these newly emerging classes is tough, and obtaining extensive labeled data for each class is even harder. The annotation process can be extremely time-consuming and expensive (Ding et al., 2022; Guo et al., 2021; Wang et al., 2022c). Naturally, it becomes crucial to empower models to classify the nodes from: limited labeled classes and those unseen classes having no labeled instances, collectively referred to as *weakly supervised*. In this regard, we investigate the problem of *Graph Class Incremental Learning under Weak Supervision (GCL)*.

Recent studies (Lu et al., 2022; Tan et al., 2022), have delved into a specific aspect of the broader problem, termed graph few-shot class incremental learning (GFSCIL). This approach operates under the assumption that the base classes possess abundant labeled instances, while novel classes introduced during streaming sessions always have representations in the form of $k$-shots. Additionally, there is a separate line of research (Wang et al., 2021b; 2023b; Hanouti & Borgne, 2022), focusing on zero-shot node classification. Furthermore, addressing the issue of limited labeled data availability during base training, which impacts the model's generalizability for better node representations during finetuning, is discussed in Wang et al.. Finally, studies like (Wang et al., 2023a; Wang et al.) address few-shot node classification. Graph data, existing in a non-Euclidean space with constantly changing network structures, poses unique challenges. Unlike the progress made in class incremental learning in computer vision, incremental learning in graphs remains relatively unexplored. Therefore, for developing a framework for *GCL* the key challenges include: *(1) Can the model learn good node representations with just k-shots for base training classes during finetuning? (2) Is there a universal framework to address both the GFSCIL problem (classes in novel streams represented by k-shots) and the GCL task (including classes with no training instances)? and finally, (3) How to prevent forgetting old knowledge while learning new information?*

Sometimes, *Less is Plenty*. By heuristically sampling the neighborhood corresponding to the $k$-shot representations, our approach extends the support set for each class. These prototypes serve a crucial role in steering the orientation of both base and novel classes within the graph during streaming sessions. We adopt the popular meta-learning strategy, called episodic learning (Finn et al., 2017), which has shown great promise in few-shot learning. We propose ***Graph Orientation Through Heuristics And Meta-learning (GOTHAM)***, an incremental learning framework that effectively addresses all the aforementioned issues. Finally, the teacher-student knowledge distillation in GOTHAM prevents catastrophic forgetting. The paper is structured into seven sections, focusing on class orientation through prototypes, proposed approach, experimental analysis, and concluding remarks.

## 2 Related Work

**Continual Graph Learning**: The Continual Graph Learning Benchmark (CGLB) (ZHANG et al., 2022) categorizes tasks in evolving graph structures into Continual Graph Learning (CGL) (Wang et al., 2022a; Xu et al., 2020; Daruna et al., 2021; Ahrabian et al., 2021; Kou et al., 2020), Dynamic Graph Learning (DGL) (Galke et al., 2020; Wang et al., 2020a; Yu et al., 2018; Han et al., 2020), and Few-Shot Graph Learning (FSGL) (Zhou et al., 2019a; Guo et al., 2021; Yao et al., 2020). CGL focuses on mitigating catastrophic forgetting without relying on past data, DGL captures temporal dynamics with access to historical data, and FSGL enables rapid adaptation to new tasks using meta-learning. Our work lies at the intersection of

CGL and FSGL. We review related work on few-shot, zero-shot, and incremental learning for graph-based tasks, positioning our contributions within this broader framework.

**Few-Shot Node Classification**: Despite several advancements in applying GNNs to node classification tasks (Kipf & Welling, 2017; Veličković et al., 2018; Hamilton et al., 2017; Wang et al., 2022b), more recently, many studies (Ding et al., 2020b; Wang et al., 2021a; Zhou et al., 2019b; Wang et al.) have shown that the performance of the GNNs is severely affected when number of labeled instances are limited. Consequently, there has been a surge in interest in the area of few-shot node classification. These works are broadly categorized into two main streams: (1) Optimization based approaches (Zhou et al., 2019b; Huang & Zitnik, 2020; Liu et al., 2021; Lan et al., 2020) and (2) Metric based approaches (Wang et al., 2023a; Wang et al.; Snell et al., 2017a; Yao et al., 2020). These approaches operate under the strong assumption that information for all classes is available simultaneously, which renders them *ineffective for class incremental learning scenarios.*

**Zero-Shot Classification**: As emerging classes continue to grow in dynamic environments, interest in a related field called "no-data learning" is surging. However, the existing approaches (Wang et al., 2021b; 2023b; Hanouti & Borgne, 2022; Lu et al., 2018; Wan et al., 2019b; Song et al., 2018) suffer from two key limitations: (1) Many of these methods assume access to unlabeled instances of unseen classes during training, limiting their generalizability and (2) They typically only classify test instances into the set of unseen classes, which isn't practical. In computer vision (Verma et al., 2019; Wu et al., 2023), some approaches have addressed these issues and even integrated incremental learning successfully. However, *similar advancements in the graph domain are lacking.*

**Class Incremental Learning**: also known as lifelong learning has been extensively studied across various computer-vision tasks (Li & Hoiem, 2018; Rebuffi et al., 2016; Hou et al., 2019). However, these approaches often assume access to extensive labeled datasets during streaming sessions, which is impractical. Few-shot class incremental learning (FSCIL) has been introduced in the realms of image classification in (Tao et al., 2020; Cheraghian et al., 2021). Unlike images, graph data exhibits non-i.i.d characteristics, making incremental learning more challenging. Most recent works (Lu et al., 2022; Tan et al., 2022) have addressed the graph few-shot class incremental learning framework. However, a common but naive assumption in these approaches is the *abundant availability of base classes, which often isn't the case in practice.* Our proposed framework aims to bridge the gap by directly addressing the limitations found in various existing works.

## 3 Methodology

In this section, we begin by presenting the problem and explaining the key terms related to it. Then, we introduce some foundational concepts that will help build our formulation. Finally, we outline several crucial modules and provide detailed explanations for each.

### 3.1 Problem Statement

We denote an attributed graph as $G^t(V^t, E^t, X^t)$, where $V^t = \{v_1^t, v_2^t, \ldots, v_n^t\}$ is the vertex set and $E^t \subseteq V^t \times V^t$ is the edge set. $X^t = \{x_1, x_2, \ldots, x_{|V^t|}\} \in \mathbb{R}^{|V^t| \times d}$, is the node feature matrix where $d$ is the feature dimension. In the base training stage, we have a base graph $G^{base}$ with $|C^{base}|$ number of classes. Due to weak supervision, the number of labeled samples corresponding to $C^{base}$ is extremely limited. In the streaming sessions, evolving graphs are presented $\{G^1, G^2, \ldots, G^T\}$ with $\{C^1, C^2, \ldots, C^T\}$ sets of classes. In the GFSCIL framework, every streaming session introduces $\delta C^i$ new classes, each represented by $k$-shots. It is essential to note that $\delta C^i \cap \delta C^j = \emptyset$ and $C^t = C^{base} + \sum_{i=1}^t \delta C^i$.

**Problem definition:** *Graph Class Incremental Learning under weak supervision*

In each streaming session, $G^t$ introduces $\delta C^i$ new classes, which are divided into two categories: $\delta C^{i,f}$ and $\delta C^{i,z}$. $\delta C^{i,f}$ classes, termed as seen classes, have few training instances (typically $k$-shots), while $\delta C^{i,z}$ classes, referred to as unseen classes lack any training instances. During the streaming session, we encounter both $\delta C^{i,f}$ and $\delta C^{i,z}$ classes, forming the class set denoted as $C^t = C^{t,S} \cup C^{t,U}$, where $S$ stands for seen classes and $U$ denotes unseen classes at time "t". Specifically, $C^{t,S} = C^{base} + \sum_{i=1}^t \delta C^{i,f}$ and $C^{t,U} = \sum_{i=1}^t \delta C^{i,z}$. Additional information, in terms of class semantics descriptions (CSDs), is provided for all the classes. The

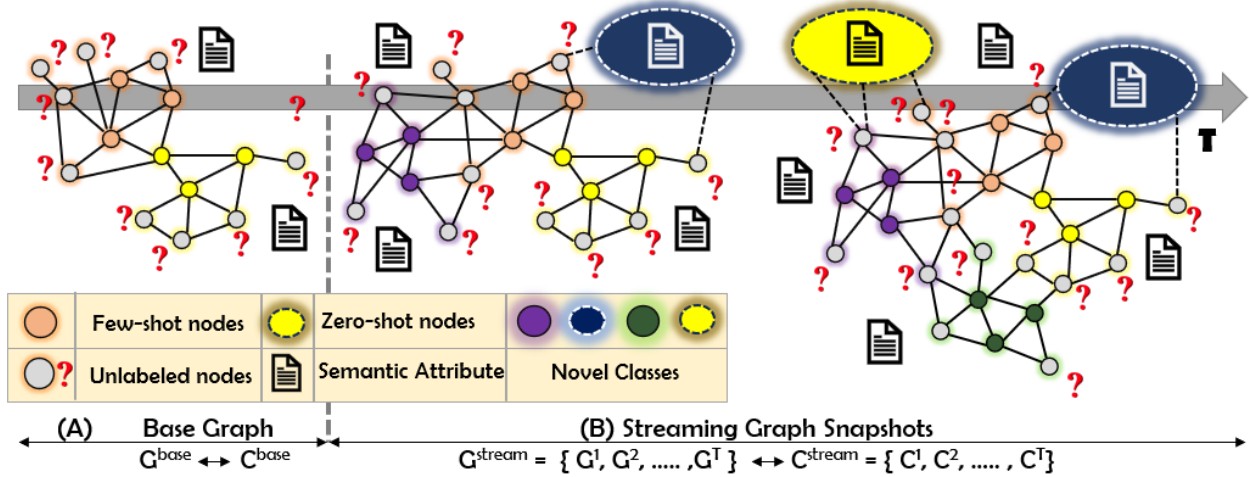

Figure 1: **Graph Class Incremental Learning under Weak Supervision**:(A) In the base graph $G^{base}$, the base classes $C^{base}$ have extremely limited labeled instances.(B) In the streaming sessions, graph $G^t$ has $C^t$ number of classes. Depending upon the availability of the training instances, the classes are further classified as $C^{t,S}$ (seen classes) and $C^{t,U}$ (unseen classes). Seen classes are represented with $k$-shots, along with semantic attributes (CSDs). For unseen classes, only CSD information is available. The goal, is to classify the unlabeled instances into $C^t$ classes encountered so far (Lu et al., 2022; Tan et al., 2022).

CSD matrix is denoted as $A_s = \{a_{s1}, a_{s2}, \ldots, a_{sC^t}\} = A_s^S \cup A_s^U$, with each row containing description of a class. Class semantics descriptions (CSDs) have been extensively studied in Wang et al. (2023b); Hanouti & Borgne (2022); Wang et al. (2021b); Ju et al. (2023). Throughout the paper, we interchangeably refer to "class semantics descriptions (CSDs)" and "semantic attributes". The goal is to classify all the unlabeled nodes (belonging to both seen and unseen classes) into $C^t$ classes encountered so far.

Labeled training instances are called "support sets" ($\mathcal{S}$), while unlabeled testing instances are termed "query sets" ($\mathcal{Q}$). Unseen classes, which lack training instances, have their unlabeled instances presented only during inference.

### 3.2 Preliminaries: Label smoothness and Poisson Learning (Random walk perspective)

To enhance classification accuracy in scenarios with extremely low labeled data, leveraging additional samples is crucial. Semi-supervised learning, which combines labeled and unlabeled data, has shown significant improvements by utilizing the topological structure of the data (Zhu et al., 2003; Zhou & Schölkopf, 2004; Zhou et al., 2003). Methods such as Poisson learning (Calder et al., 2020) have further extended the concept by incorporating structure-based information on graphs through random walks. The underlying assumption is that samples that are close to each other can potentially share similar classes. Previous research (Solomon et al., 2014; Belkin et al., 2006; Kalofolias, 2016), has emphasized the importance of the smoothness assumption for label propagation in scenarios with extremely low label rates. These findings form the basis of our proposed approach, which extends support sets through random walks without requiring extensive labeled nodes. This extended support set enhances the representation of prototypes for each class, leading to improved classification performance.

### 3.3 Prototype representation

*Definition*: The prototype of a class corresponds to a representative embedding vector which captures the overall characteristics of a class in the attribute space. Prototype representation has been extensively studied across various works (Snell et al., 2017b; Rebuffi et al., 2017; Lu et al., 2022; Tan et al., 2022) in the domain of few-shot representation learning.

The foundational works, (Snell et al., 2017b; Rebuffi et al., 2017) suggested using the mean of the support samples for prototype representation. Building upon this foundation, subsequent studies (Lu et al., 2022; Tan et al., 2022) introduced attention-based prototype generation techniques. These approaches were designed to address challenges such as class imbalance and mitigate biases arising from noisy support sets. Recent studies (Wang et al.; 2023a) highlighted the importance of neighborhood sampling techniques, such as Poisson learning and personalized page rank (PPR), to obtain a more informed support set.

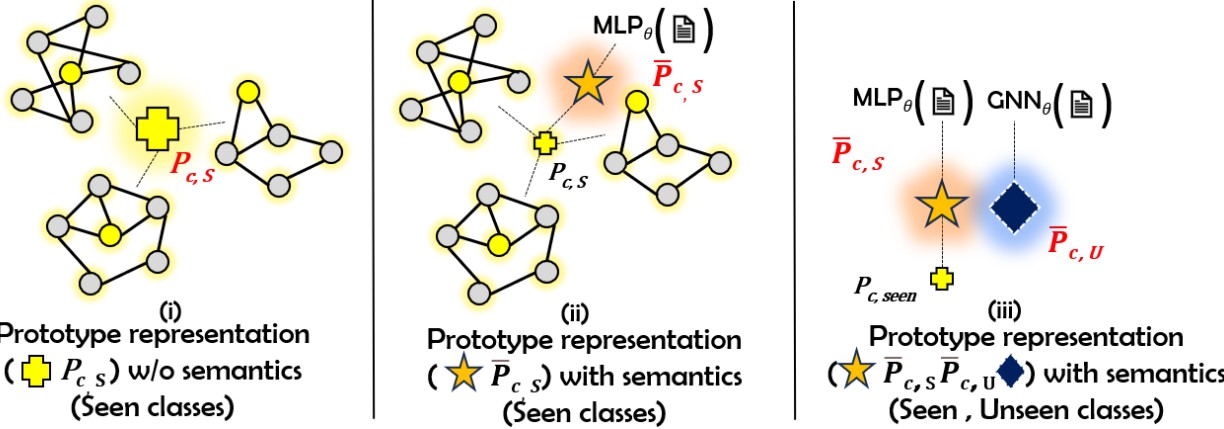

Figure 2: **Prototype representation**: For the GFSCIL task, we propose representing prototypes ($P_{c,S}$) using the averaged extended support set, as illustrated in (i). As demonstrated in (ii), we integrate semantic attributes (CSDs) to enhance the prototypes ($\overline{P}_{c,S}$) in TAGs. For GCL tasks with classes having no training instances, the semantic attributes (CSDs) are encoded as prototypes ($\overline{P}_{c,U}$).

However, these approaches (Snell et al., 2017b; Rebuffi et al., 2017; Lu et al., 2022; Tan et al., 2022; Wang et al.; 2023a) suffer due to the weak supervision setting. Hence, we additionally leverage the label smoothness principle to gather the local neighborhood of the support nodes. The extended support set contains the labeled support nodes and the unlabeled neighbors gathered through random walks. The final node set for a class can be represented as $\mathcal{S}_{x,C} = \mathcal{S}_C \cup \mathcal{V}_C$, where $\mathcal{S}_C$, corresponds to nodes with labels and $\mathcal{V}_C$ is the sampled unlabeled node set for class $C$ belonging to the seen classes. The prototype thus obtained will be the average of the embeddings of all the nodes within the extended support set $\mathcal{S}_{x,C}$ represented as:

$$P_{C,S} := \frac{1}{|\mathcal{S}_{x,C}|} \sum_{i=1}^{|\mathcal{S}_{x,C}|} \mathrm{GNN}_\theta (v_i) \tag{1}$$

where, $\mathrm{GNN}_\theta (v_i)$ corresponds to embeddings of the node $v_i$, which are generated by aggregating information from its neighbors. Refer to Figure 2(i) for further details. In Text Attributed Graphs (TAGs), refer to Figure 2(ii), we enhance the prototype representation by incorporating semantic attributes associated with each class. The semantic loss, which will be elaborated upon later, facilitates the integration of attribute and semantic space. The encoded semantic attributes $\mathrm{MLP}_\phi (a_{sC})$, for the same seen class $C$ are merged with the original prototype to obtain the *new prototype representation*: $\bar{P}_{C,S} = \frac{P_{C,S} + \mathrm{MLP}_\phi(a_{sC})}{2}$. For the GFSCIL framework, the prototype set will consist of $P_S = \{P_{C,S}, \bar{P}_{C,S}\}$, depending on the type of graph used as input. For a class $\hat{C}$, with no training examples (where $\hat{C} \in C^{t,U}$), we rely on additional information in the form of semantic attributes. Refer to Figure 2(iii) for further insights into this representation. The prototype representation for such unseen classes is: $\bar{P}_{\hat{C},U} = \mathrm{GNN}_\theta (a_{s\hat{C}})$, where $\mathrm{GNN}_\theta (a_{s\hat{C}})$, represents the vector representation of the semantic attributes, where each attribute connects only to itself (self-loop) in its adjacency. In the GCL scenario, the prototype representation set is denoted as $\bar{P} = \{\bar{P}_{C,S}, \bar{P}_{\hat{C},U}\}$.

## 3.4 Transferable Metric Space Learning

Graphs constantly change, posing a challenge in how different node classes are positioned in metric space. As information for all classes isn't available at once, it is crucial to establish a criteria to prevent overlap

between old and novel classes. Additionally, distinguishing novel classes as seen or unseen adds another layer of complexity, especially when some classes have very few samples while others appear only during inference. To address these challenges, the model is trained using the following loss functions:

**Intra-class clustering loss**: The goal here is to group instances that belong to the same class together. Several data augmentation strategies have been suggested previously (Li et al., 2018; Verma et al., 2020; Ding et al., 2018; Qiu et al., 2020), which generate consistent samples without affecting the semantic label. However, in few-shot scenarios, these methods can introduce bias by generating samples that do not fully capture the true distribution of the class. To avoid this, we take a simpler approach: sampling the neighborhood of the original node to obtain a correlated view. In our case, the original nodes correspond to the labeled k-shot representative nodes for each seen class. Neighbor nodes are sampled with the approach discussed in section: 3.2. The class prototype is responsible for grouping these nodes, employing the clustering loss defined as:

$$L_{cls,S} := \frac{1}{|C^{t,S}|} \sum_{j \in C^{t,S}} \left( \sum_{i \in n_j} \frac{\max(\|\text{GNN}_\theta(v_i) - P_{j,S}\| - \gamma, 0)}{\sum_{k \in n_j} \max(\|\text{GNN}_\theta(v_k) - P_{j,S}\| - \gamma, 0)} \right) \tag{2}$$

Here, $C^{t,S}$ refers to the set of seen classes encountered up to the current streaming session at a time "$t$". The number of samples for each class from the extended support set is represented by "$n_j$". The parameter "$\gamma$" defines the boundary from the prototype. Samples for a certain class are encouraged to stay within this boundary. The max(.) function ensures that only samples outside the boundary contribute to the loss.

**Inter-class segregation loss**: Unlike existing augmentation strategies (Li et al., 2018; Verma et al., 2020; Ding et al., 2018; Qiu et al., 2020), which generate correlated pairs within a class, we do not rely on explicitly sampling negative pairs. Instead, we promote class separability by leveraging representative embedding vectors, known as class prototypes. These prototypes enhance dissimilarity between samples from different classes, preventing class overlap among all classes ($C^t$) encountered up to the current streaming session at time "$t$". The segregation loss is defined as follows:

$$L_{seg} := \frac{-1}{|C^t|} \sum_{j \in C^t} \sum_{p \in C^t, p \neq j} \log\|\bar{P}_j - \bar{P}_p\| \tag{3}$$

Here, $C^t = \{C^{t,S} \cup C^{t,U}\}$ is the set of all classes, including both seen and unseen classes. Similarly, the prototypes belong to the prototype representation set $\bar{P} = \{\bar{P}_{C,S}, \bar{P}_{\hat{C},U}\}$. This loss function applies to both seen and unseen classes.

**Semantic manipulation loss**: Each modality offers a unique perspective on class representation, contributing to a more comprehensive view of prototypes. While extensively explored in the image domain (Zhang et al., 2023; Xing et al., 2019; Xu & Le, 2022; Guan et al., 2021), graphs provide an additional advantage by incorporating structural information (orientation) associated with each class within the graph. Class-semantic descriptors (CSDs) or semantic attributes, derived from class names and descriptions, are encoded and represented as $\text{MLP}_\phi(a_{sC})$ for $C \in C^{t,S}$. The objective is to align the encoded semantics with the prototypes of the seen classes. The corresponding loss function is expressed as follows:

$$L_{sem,S} := \sum_{j \in C^{t,S}} \|\text{MLP}_\phi(a_{sj}) - P_{j,S}\| \tag{4}$$

This loss function is responsible for integrating the attribute and semantic space. The newly learned semantic embeddings are later merged to obtain a new prototype representation (discussed previously). This loss function is specifically applied only to seen classes.

**Knowledge refinement through experience**: As the graph evolves incrementally, the learner model may tend to forget previously learned information when exposed to new knowledge, leading to catastrophic forgetting. To address this, it's crucial to preserve the previously acquired knowledge while integrating new information. This process is known as knowledge distillation. Among various techniques (Zhang et al., 2020; Rezayi et al., 2021; Feng et al., 2022), we opt for the teacher-student approach. The teacher model distills

both attribute and semantic information using the following loss function:

$$L_{emb,S} = \frac{1}{n_{C^{(t-1),S}}} \sum_{i \in n_{C^{(t-1),S}}} \left\| \text{GNN}_\theta^{teacher}(v_i) - \text{GNN}_\theta^{student}(v_i) \right\| \tag{5}$$

$$L_{align,S} = \frac{1}{|C^{(t-1),S}|} \sum_{j \in C^{(t-1),S}} \left(1 - \frac{\text{MLP}_\phi^{teacher}(a_{sj}) \cdot \text{MLP}_\phi^{student}(a_{sj})}{\left\|\text{MLP}_\phi^{teacher}(a_{sj})\right\| \left\|\text{MLP}_\phi^{student}(a_{sj})\right\|}\right) \tag{6}$$

The total loss is: $L_{KD,S} := \lambda_1 \cdot L_{emb,S} + \lambda_2 \cdot L_{align,S}$. For the GFSCIL problem, where text attributes are not available, knowledge distillation is solely performed across the node-embeddings (attribute information). This loss function applies only to seen classes.

## 4 Understanding Prototype Distortion in Evolving Graphs

Although we introduce various components to maintain model performance in a continual learning setup with class increments, the key question remains: *Can prototype representations truly be preserved across evolving data streams?* The answer depends on how well GNNs can express and adapt these prototypes as the graph evolves. In this subsection, we evaluate the stability of prototype representations within an evolving graph.

GNNs suffer from representation distortion over time Liang et al. (2018); Wu et al. (2022); Lu et al. (2023), leading to gradual performance degradation. This shift can result from the continuous addition of nodes and edges, structural changes, new feature introductions, or the emergence of new classes. Additionally, the lack of labeled data further increases the challenge. As shown in Figure 5, our experimental evaluations confirm this trend—model performance steadily declines as novel classes are introduced over incremental stages. To better understand this degradation, we build on insights from Lu et al. (2024), and extend the concept of representation distortion to prototype distortion in our case. Before presenting the results, we first outline the following assumptions:

**Assumptions:** We consider an initial graph $G^{base} = (V^0, E^0, X^0)$ with $n$ nodes. (1) The feature matrix $X^0$ follows a continuous probability distribution over $\mathbb{R}^{n \times d}$. (2) At each time step $t$, new nodes belonging to classes indexed as $\delta C^t$ are added, connecting to existing nodes with a positive probability while preserving existing edges. In citation networks, for instance, the new papers cite existing ones, maintaining stable relationships. Over time, emerging research fields introduce papers from previously unseen domains. (3) The feature matrix $X^{\delta C^t}$ has a zero mean conditioned on prior graph states, i.e., $E[X^{\delta C^t}|G^0, ..., G^{t-1}] = 0_d, \quad \forall t \geq 1$ This follows a common assumption in deep learning models.

We specifically examine the prototypes corresponding to the classes in $\Delta C = C^t \cap C^{t+1}$, representing the shared classes between consecutive time steps obtained through the GNN parameterized by $\theta$. We define the expected distortion for prototype $P_i$ at time $t$ as the expected difference between its representation at time $t$ and $t+1$, given by: $\Delta(P_i, \delta t) = \mathbb{E}\left[\|P_i(t+1) - P_i(t)\|^2\right]$, where $\Delta(P_i, \delta t)$ denotes the expected distortion of prototype $P_i$ over the time interval $\delta t$.

**Theorem:** *If $\theta$ represents the vectorized parameter set $\{(a_j, W_j, b_j)\}_{j=1}^N$, where each coordinate $\theta_i$ is drawn from the uniform distribution $U(\theta_i^*, \xi)$ centered at $\theta_i^*$ (optimal parameters), the expected deviation $\Delta(P_i, \delta t)$ due to the perturbed GNN model at time $t \geq 0$ for prototype $P_i$, where $i \in \Delta C$, is lower bounded by:*

$$\Delta(P_i, \delta t) \geq \mathbb{E}\left[\left(\frac{1}{d_{t+1}(P_i)} - \frac{1}{d_t(P_i)}\right)^2 \sum_{k \in N_t(P_i)} \|x_k\|^2\right] \tag{7}$$

where the set $N_t(P_i)$, denotes the neighborhood comprising the extended support set of the prototype $P_i$ and $d_t(P_i), d_{t+1}(P_i)$ refers to degree information at respective time steps. The proof of the theorem can be found in the ***Appendix***.

**Remark:** Under the ever-growing assumption, distortion is unavoidable, and Theorem confirms that the expected distortion of the model output increases strictly over time. This effect is particularly pronounced for large-width models, emphasizing the inherent trade-off in continuously evolving systems.

# 5 Proposed Algorithm

In this section, we present our proposed framework: **Graph Orientation Through Heuristics And Meta-learning (GOTHAM)**. At any given time $t$, we have a graph $G^t$ as input, where the total classes $C^t = C^{t,S} \cup C^{t,U}$ encompass both the seen (few shots) and the unseen (zero shots) class representation. The choice of framework type depends on the input, as illustrated in Figure 3. Based on the input, the following procedure is used to perform the node classification:

**(1) Episodic Learning:** Based on the choice of framework, tasks ($\mathcal{T}$) are sampled for the corresponding graph $G^t$. Each task $\mathcal{T}^i \sim p(\mathcal{T})$, drawn from the task distribution, consists of an extended support set ($\mathcal{S}_x^i$) and a query set ($\mathcal{Q}^i$) required for episodic learning. Episodic learning, which has demonstrated great promise in the area of few-shot learning (Rebuffi et al., 2017; Tan et al., 2022; Huang & Zitnik, 2020; Vinyals et al., 2016; Zhou et al., 2019b), involves sampling tasks and learning from them, rather than directly training and then fine-tuning over batches of data. **(2) Prototype representation:** For each support set ($\mathcal{S}_x^i$), prototypes are generated for all the classes. If the class set $C^t$ contains samples only from the seen classes (i.e. $C^t = C^{t,S}$), the prototype set will be $P_S$ and the problem becomes a GFSCIL setting. Furthermore, if TAGs are given as an input, which offer additional semantic attribute information, it results in a new prototype representation. For the GCL setting where $C^t = C^{t,S} \cup C^{t,U}$, the prototype representation set is denoted as $\bar{P} = \{\bar{P}_{C,S}, \bar{P}_{\hat{C},U}\}$. **(3) Meta-learning and Finetuning**: After obtaining prototypes, meta-training is performed using a combination of loss functions. In the GFSCIL scenario without TAGs, the model is trained using clustering loss and separability loss. With TAGs in both GFSCIL and GCL settings, semantic loss is also incorporated. The corresponding meta-training loss is defined as: $L_{train} := \alpha_1 \cdot L_{cls,S} + \alpha_2 \cdot L_{seg} + \alpha_3 \cdot L_{sem,S}$. Meta-learning is performed on the base graph. Once the model is trained on the support set ($\mathcal{S}_x^i$), its performance is validated on the corresponding query set ($\mathcal{Q}^i$). The model is then frozen and used for meta-finetuning. During meta-finetuning, the loss is defined as: $L_{finetune} := \alpha_1 \cdot L_{cls,S} + \alpha_2 \cdot L_{seg} + \alpha_3 \cdot L_{sem,S} + \alpha_4 \cdot L_{KD,S}$. **(4) Knowledge distillation:** During finetuning, knowledge distillation preserves previously learned class representations. The corresponding loss is defined as: $L_{KD,S} := \lambda_1 \cdot L_{emb,S} + \lambda_2 \cdot L_{align,S}$. **(5) Node classification:** For any graph $G^t$ as input at time $t$, the model ultimately performs $C^t$-way node classification.

---

**Algorithm 1 GOTHAM III.o**

---

1: **Input:** $G^t, A_s, C^t$
2: **Output:** Label prediction on query nodes in $\mathcal{Q}^i \in \mathcal{T}^i$
3: # Initialise $\theta, \phi$; Sample $\mathcal{T}^i \sim p(\mathcal{T})$
4: **Base Training** ($t = 0, C^t = C^{base}$)
5:     **for** $\mathcal{T}^i = \{\mathcal{S}_x^i \cup \mathcal{Q}^i\} \in C^{base}$:
6:         Prototype ($\bar{P}_{C^{base},S}$) using eq:1
7:         Compute $L_{train}$ on $\mathcal{S}_x^i$
8:         Obtain node labels for $\mathcal{Q}^i$
9:         Update $\theta, \phi$ using gradient descent
10:     **end**
11:     # Freeze the trained model
12: **Finetuning** ($t > 0, C^t = C^{base} + \sum_{i=1}^{t} \delta C^i$)
13:     # Load pre-trained model; Perform knowledge distillation
14:     **for** $\mathcal{T}^i = \{\mathcal{S}_x^i \cup \mathcal{Q}^i\} \in C^t$:
15:         Prototype ($\bar{P}_{C^t}$) using eq:1
16:         Compute $L_{finetune}$ on $\mathcal{S}_x^i$
17:         Obtain node labels for $\mathcal{Q}^i$
18:         Update $\theta, \phi$ using gradient descent
19:     **end**

---

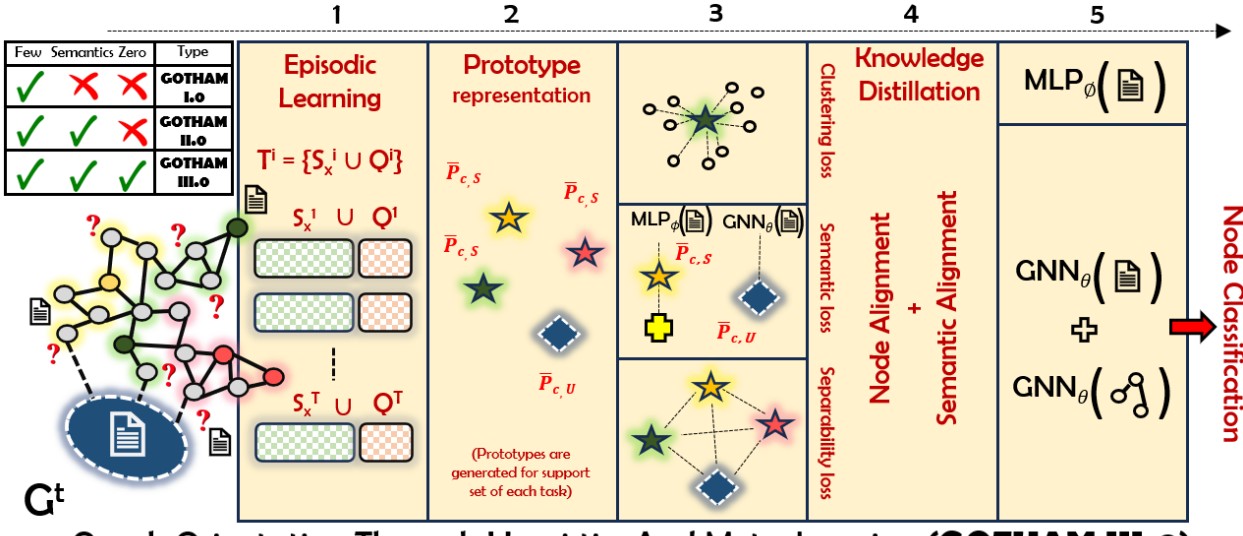

Figure 3: **GOTHAM III.o**: At any time $t$, the framework uses the graph $G^t$ as input. The total classes are $C^t = C^{t,S} \cup C^{t,U}$. The steps are: (1) Create tasks ($T$) with support sets ($\mathcal{S}$) and query sets ($\mathcal{Q}$) for episodic learning. (2) Obtain prototype representations for each support set ($\mathcal{S}_x^i$). (3) Apply loss functions. (4) Use knowledge distillation to transfer knowledge from the teacher model to the student model. (5) Perform node classification.

## 6 Experiments

*Datasets:* We assess the performance of our proposed framework, GOTHAM, on three real-world datasets-Cora-ML, Amazon, and OBGN-Arxiv. We summarize the statistics of the datasets in Table 1. For more details about the dataset refer to the ***Appendix***.

Table 1: Statistics of datasets used in the experiments

| Dataset | Nodes | Features | Classes | Class Labels | Tasks |
|---|---|---|---|---|---|
| Cora-ML | 2,708 | 1,433 | 7 | Neural Network, Rule Learning, Reinforcement Learning, Probabilistic Methods, Theory, Genetic Algorithms, Case-based | GFSCIL, GCL |
| Amazon | 13,752 | 767 | 10 | Label names Unavailable | GFSCIL |
| OBGN-Arxiv | 169,343 | 128 | 40 | Arxiv cs na, Arxiv cs mm, Arxiv cs lo, Arxiv cs cy, Arxiv cs cr, Arxiv cs dc, Arxiv cs hc, Arxiv cs cv, Arxiv cs ai, ... | GFSCIL, GCL |

*Experiment settings:* We partition the dataset into base stage and multiple streaming sessions respectively. We assess our framework across two main problem settings: (1) Graph Few-shot Class Incremental Learning (GFSCIL) and (2) Graph Few-shot Class Incremental Learning under Weak Supervision (GCL). Cora-ML and OBGN-Arxiv are Text-Attributed Graphs (TAGs), enriched with semantic attributes. We generate semantic attributes/ Class Semantics Descriptors (CSDs) using "word2vec" (Mikolov et al., 2013), which transform textual descriptors into word embeddings. To simplify computation, we utilize Label-CSDs (Wang et al., 2021b). Initially, we evaluate all datasets under the GFSCIL setting. For the Cora-ML and Amazon dataset, we choose five classes as the novel classes and keep the rest as base classes, and adopt *1-way, 5-shot* setting, which means we have 6 sessions (1 base sessions + 5 novel sessions). For the OBGN-Arxiv dataset, we keep ten classes as base classes and the rest as novel, employing a *3-way, 10-shot* setting (totaling 11 sessions). Our framework seamlessly integrates semantic attribute information in Cora-ML and OBGN-Arxiv. Finally, we assess our framework under the GCL setting, focusing on Cora-ML and OBGN-Arxiv to demonstrate its

effectiveness. During each streaming session, one class is designated as zero-shot, lacking training instances. Unlabeled instances for these classes are only available during inference. All the experiments are performed five times to ensure reproducibility. The top results are highlighted in **bold**, while the second best ones are underlined.

*Baseline methods:* In the GFSCIL setting, we benchmark our results against several state-of-the-art frameworks for few-shot class incremental learning and few-shot node classification, including: Meta-GNN (Zhou et al., 2019b), GPN (Ding et al., 2020a), iCaRL (Rebuffi et al., 2017), HAG-Meta (Tan et al., 2022), Geometer (Lu et al., 2022) and CPCA (Ren et al., 2023). Unlike previous methods, during base training, we provide only a limited number (*5- shots for Cora-ML and Amazon and 10-shots for OBGN-Arxiv*) of labeled instances for each class. In streaming sessions, novel classes receive $k$-shot representations. In the GCL setting, where novel classes have both few-shot and zero-shot representations, we compare against zero-shot learning frameworks with inductive learning as baselines. These approaches include DCDFL (741, 2024), GraphCEN (Ju et al., 2023), (CDVSc, BMVSc, WDVSc) (Wan et al., 2019a) and Random guess, introduced as a naive baseline. Unlike the traditional approach, the seen classes have only limited labeled instances (*5-shots for Cora-ML and 10-shots for OBGN-Arxiv*) available for training, and unseen classes have semantic attributes only. Under the GCL setting, the unlabeled instances will be classified into $C^t$ classes encountered, resembling a *generalized zero-shot with inductive learning framework*. A detailed summary of the baseline methods and hyper-parameters employed is available in the ***Appendix***.

**Graph Few-shot Class Incremental Learning (GFSCIL)**: We conducted experiments on the Amazon dataset, focusing on the GFSCIL problem as previously described. The results, detailed in Table 2, show an average improvement of around 6% across various streams. Visualizing the class prototypes generated by the GOTHAM framework reveals distinct separations among classes across streams, ensuring consistent performance.

Table 2: **GFSCIL setting:** (**Left**) Model performance (%) on the Amazon dataset under GFSCIL setting. (**Right**) Visualization of class prototypes for the Amazon dataset across different streaming sessions.

| Amazon (1-way 5-shot GFSCIL setting) | | | | | | |
|---|---|---|---|---|---|---|
| Stream | Base | $S_1$ | $S_2$ | $S_3$ | $S_4$ | $S_5$ |
| Meta-GNN | **99.60** | 86.33 | 82.43 | 77.75 | 70.82 | 67.94 |
| GPN | 93.56 | 85.23 | 74.88 | 73.40 | 66.17 | 63.36 |
| iCaRL | 66.20 | 47.33 | 39.13 | 35.75 | 29.84 | 29.66 |
| HAG-Meta | 95.43 | 88.76 | 75.67 | 69.56 | 67.21 | 61.86 |
| GEOMETER | 95.44 | 90.05 | 77.36 | 74.27 | 73.08 | **74.36** |
| CPCA | 95.37 | 87.88 | 83.72 | 77.13 | 76.37 | 69.32 |
| **GOTHAM I.o** | 96.61 | **90.91** | **88.89** | **84.55** | **78.82** | 73.81 |
| **%gain** | **-03.00** | **00.00** | **06.17** | **08.74** | **03.20** | **00.00** |

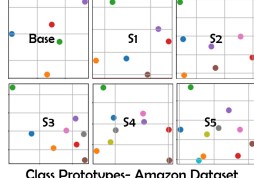

Class Prototypes- Amazon Dataset

We extend our experiments to the Cora-ML and OBGN-Arxiv datasets, both Text-Attributed Graphs (TAGs) enriched with semantic attributes. Following the previously outlined experimental conditions, GOTHAM achieved an average improvement ranging from 6.4% to 13.5% over the baseline methods. We explored two variants of the framework: GOTHAM I.o, which solely relies on feature-based information, and GOTHAM II.o, which integrates semantic attributes. Table 3 demonstrates that incorporating semantic attributes in GOTHAM II.o notably enhances performance for both datasets.

Table 3: **GFSCIL with semantics:** Node classification accuracy (%) in the GFSCIL setting- leveraging semantic attributes for enhanced class representation on Cora-ML and OBGN-Arxiv datasets with GOTHAM.

| Cora-ML (1-way 5-shot GFSCIL setting) | | | | | | |
|---|---|---|---|---|---|---|
| Stream | Base | $S_1$ | $S_2$ | $S_3$ | $S_4$ | $S_5$ |
| Meta-GNN | 100 | 79.19 | 61.37 | 60.40 | 51.51 | 36.76 |
| GPN | 95.58 | **91.89** | 77.95 | 68.57 | 70.53 | 62.53 |
| iCaRL | 93.00 | 69.13 | 53.81 | 47.20 | 42.86 | 38.60 |
| HAG-Meta | 96.08 | 87.81 | 73.96 | 70.12 | 66.19 | 60.17 |
| GEOMETER | 96.46 | 89.91 | 77.58 | 70.20 | 54.50 | 62.76 |
| CPCA | 97.67 | 90.68 | 77.38 | 75.38 | 69.50 | 59.86 |
| **GOTHAM I.o** | 100 | 90.15 | 87.83 | 83.66 | 76.56 | **75.03** |
| **GOTHAM II.o** | 100 | 91.43 | **88.69** | **84.00** | **76.92** | 72.40 |
| **%gain** | **00.00** | **00.00** | **13.78** | **11.43** | **09.06** | **19.55** |

| OBGN-Arxiv (3-way 10-shot GFSCIL setting) | | | | | | |
|---|---|---|---|---|---|---|
| Stream | Base | $S_1$ | $S_2$ | $S_6$ | $S_9$ | $S_{10}$ |
| Meta-GNN | 76.60 | 66.10 | 57.38 | 36.55 | 29.77 | 28.82 |
| GPN | 78.38 | 68.21 | 57.88 | 35.77 | 28.78 | 30.12 |
| iCaRL | 62.80 | 39.54 | 35.22 | 21.97 | 15.68 | 16.45 |
| HAG-Meta | 77.17 | 68.19 | 58.22 | 37.13 | 28.28 | 24.68 |
| GEOMETER | 80.08 | **70.68** | **61.07** | 38.13 | 29.65 | 26.22 |
| CPCA | 69.71 | 56.96 | 50.39 | 33.35 | 25.76 | 24.88 |
| **GOTHAM I.o** | 72.33 | 59.94 | 47.84 | 30.31 | 25.12 | 25.44 |
| **GOTHAM II.o** | **82.91** | 70.20 | 60.26 | **40.53** | **31.38** | **32.38** |
| **%gain** | **03.53** | **00.00** | **00.00** | **06.29** | **05.41** | **07.50** |

**Graph Class Incremental Learning under Weak Supervision (GCL)**: In a broader problem setting where novel classes have both few-shot and zero-shot representation, we conducted extensive experiments

on the Cora-ML and OBGN-Arxiv datasets. The results in Table 4 indicate an average improvement of 7% to 54% across various streams over the baselines, showcasing the effectiveness of our framework.

Table 4: **GCL setting:** Node classification accuracy (%) on the OBGN-Arxiv dataset under the GCL setting. In each streaming session, one class is designated as zero-shot, lacking any training examples.

| OBGN-Arxiv (2-way 10-shot, 1-way 0-shot GCL setting) | | | | | | | | | | |
|---|---|---|---|---|---|---|---|---|---|---|
| Stream | Base | $S_1$ | $S_2$ | $S_3$ | $S_4$ | $S_5$ | $S_6$ | $S_7$ | $S_8$ | $S_9$ | $S_{10}$ |
| Random guess | 18.10 | 14.62 | 12.50 | 09.47 | 08.64 | 08.00 | 06.43 | 05.81 | 05.59 | 04.86 | 05.00 |
| CDVSc | 68.35 | 50.86 | 43.02 | 37.68 | 29.28 | 26.03 | 22.12 | 19.78 | 15.33 | 11.48 | 10.73 |
| BMVSc | 68.38 | 51.54 | 44.28 | 36.78 | 30.30 | 27.22 | 21.98 | 20.12 | 19.78 | 10.56 | 09.34 |
| WDVSc | 67.22 | 50.98 | 45.02 | 35.78 | 29.54 | 26.77 | 21.67 | 18.34 | 16.88 | 11.56 | 07.86 |
| GraphCEN | 77.13 | 62.37 | 51.76 | 38.42 | 28.92 | 18.80 | 15.36 | 10.56 | 08.92 | 08.56 | 06.53 |
| DCDFL | 64.66 | 53.42 | 45.06 | 32.76 | 30.48 | 30.57 | 28.44 | 25.89 | 24.18 | 22.00 | 19.62 |
| GOTHAM III.o | 82.91 | 68.11 | 59.90 | 47.68 | 43.67 | 43.31 | 38.76 | 36.41 | 34.62 | 32.55 | 30.28 |
| %gain | 07.49 | 09.20 | 15.73 | 24.10 | 43.27 | 41.67 | 36.29 | 40.63 | 43.20 | 47.95 | 54.33 |

**Ablation Study:** We conducted a detailed analysis of our framework across three different aspects: **(A) Contribution of different loss functions:** Various loss functions contribute differently to optimal model performance. For this analysis, we selected the Cora-ML dataset, and the corresponding plot is available in Figure 4 (A). **(B) Support set sampling:** We categorized the dataset into small, moderate, and large-sized graphs. To ensure generalizability across other datasets, we performed support set sampling using k-hop random walks, with the ideal hop length observed between 2-4 hops from the labeled nodes. Refer to Figure 4 (B) for more details. **(C) GNN backbones:** We examined the role of different GNN architectures within the GOTHAM framework for the Cora-ML and Amazon datasets. Interestingly, performance remained consistent across different architectures, indicating model-agnostic behavior. Refer to Figure 4 (C) for more details.

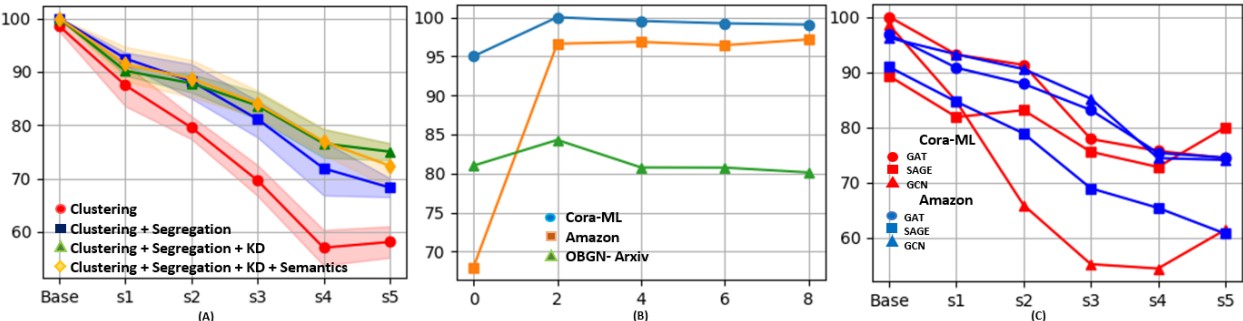

Figure 4: (A) Contribution of different loss functions on the Cora-ML dataset. (B) Support set sampling: determining ideal random-walk length. (C) Different GNN backbones on Cora-ML and Amazon datasets. (A) and (C) displays performance vs streaming sessions, while (B) shows performance vs random-walk length.

Figure (5) presents a detailed analysis of the Graph Class Incremental Learning under Weak Supervision (GCL) setting, showcasing the performance of our model across different variants of GOTHAM for various tasks on the Cora-ML and OBGN-Arxiv datasets. The plots offer an overview of GOTHAM's performance across different representations encountered during few-shot and zero-shot learning scenarios. Notably, the model maintains consistent performance even when faced with a heavy influence of unseen classes during streaming sessions. To simplify understanding: the experimental setup for base training remains consistent throughout. During streaming sessions, where we adopt an $n$-way, $k$-shot strategy, we experiment with different values of $n$ while setting $k$ to zero.

## 7 Conclusion

In this study, we introduced GOTHAM, a class incremental learning framework designed for weakly supervised settings. We initially addressed the GFSCIL problem setting, where access to labeled data during base training is limited. Our experiments highlighted the advantages of incorporating semantic attributes for Text-Attributed Graphs (TAGs). We then expanded our scope to a broader objective, Graph Class

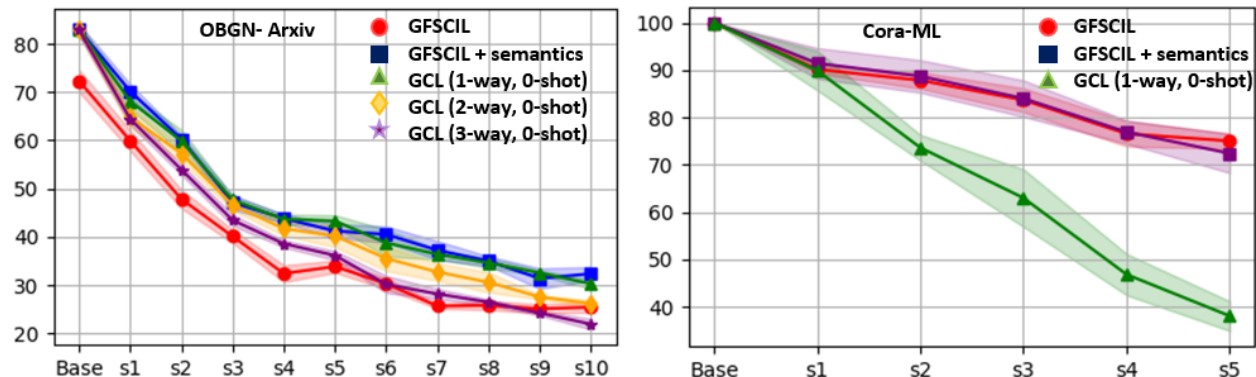

Figure 5: Performance analysis of GOTHAM framework on OBGN-Arxiv and Cora-ML datasets. **(Left)**: GCL with a 3-way $k$-shot setting shows consistent performance, even in zero-shot learning cases. **(Right)**: GCL with the 1-way $k$-shot setting on Cora-ML.

Incremental Learning under Weak Supervision (GCL), where novel classes have both a few-shot and zero-shot representation. Through extensive experiments, we conclusively established the generalizability and effectiveness of our framework across diverse tasks.

## 8 Acknowledgment

We sincerely thank the reviewers for their valuable feedback and insightful suggestions, which have greatly improved the quality of this paper. We also appreciate the support and engaging discussions from the MISN lab, led by Dr. Sandeep Kumar, and Dr. Prathosh A.P. from IISc Bangalore. This research was supported by the DST Inspire Faculty Grant (MI02322G), and Dr. Prathosh A.P. is additionally supported by the Infosys Foundation Young Investigator Award.

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

## 9 Appendix

### 9.1 Datasets

We assess the performance of our proposed framework, GOTHAM, on three real-world datasets- Cora-ML, Amazon, and OBGN-Arxiv. The detailed description is in Table 5:

Table 5: Statistics of datasets used in the experiments

| Dataset | Nodes | Features | Classes | Class Labels | Tasks |
|---------|-------|----------|---------|--------------|-------|
| Cora-ML | 2,708 | 1,433 | 7 | Neural Network, Rule Learning, Reinforcement Learning, Probabilistic Methods, Theory, Genetic Algorithms, Case-based | GFSCIL, GCL |
| Amazon | 13,752 | 767 | 10 | Label names Unavailable | GFSCIL |
| OBGN-Arxiv | 169,343 | 128 | 40 | Arxiv cs na, Arxiv cs mm, Arxiv cs lo, Arxiv cs cy, Arxiv cs cr, Arxiv cs dc, Arxiv cs hc, Arxiv cs cv, Arxiv cs ai, ... | GFSCIL, GCL |

**Cora-ML** (Bojchevski & Günnemann, 2017): This is an academic network of machine learning papers. The dataset contains 7 classes, with each node representing a paper and each edge representing a citation between papers.

**Amazon** (Hou et al., 2020): This dataset represents segments of the Amazon co-purchase e-commerce network. Each node is an item, and each edge denotes a co-purchase relationship by a common user. Node features are bag-of-words encoded product reviews, and class labels correspond to product categories.

**OBGN-Arxiv** (Subramanian et al., 2005): This dataset is a directed graph representing the citation network of Computer Science arXiv papers indexed by MAG. Each node is an arXiv paper, and each directed edge indicates a citation from one paper to another. Each paper has a 128-dimensional feature vector, created by averaging the embeddings of words in its title and abstract.

### 9.2 Baseline methods

In the GFSCIL setting, we benchmark our results against several state-of-the-art frameworks for few-shot class incremental learning and few-shot node classification, including:

#### 9.2.1 Few-shot node classification

**Meta-GNN** (Zhou et al., 2019b): Meta-GNN addresses few-shot node classification in graph meta-learning. It learns from numerous similar tasks to classify nodes from new classes with few labeled samples. Meta-GNN is versatile and can be easily integrated into any state-of-the-art GNN.

**Graph Prototypical Network (GPN)** (Ding et al., 2020a): GPN is an advanced method for few-shot node classification. It uses graph neural networks and meta-learning on attributed networks for metric-based few-shot learning.

#### 9.2.2 Class incremental learning

**Incremental classifier and representation learning (iCaRL)** (Rebuffi et al., 2017): iCaRL is a class-incremental method for image classification. We enhance it by replacing the feature extractor with a two-layer GAT network.

**Hierarchical-Attention-based Graph Meta-learning (HAG-Meta)** (Tan et al., 2022): HAG-Meta follows the graph pseudo-incremental learning approach, allowing the model to learn new classes incrementally by cyclically adopting them from base classes. It also tackles class imbalance using hierarchical attention modules.

**Graph Few-Shot Class-Incremental Learning via Prototype Representation (Geometer)**(Lu et al., 2022): Geometer predicts a node's label by finding the nearest class prototype in the metric space and adjusting the prototypes based on geometric proximity, uniformity, and separability of novel classes. To address catastrophic forgetting and unbalanced labeling, it uses teacher-student knowledge distillation and biased sampling.

**Class Prototype Construction and Augmentation (CPCA)** (Ren et al., 2023): CPCA is a method that constructs class prototypes in the embedding space to capture rich topological information of nodes or graphs, representing past data for future learning. To enhance the model's adaptability to new classes, CPCA uses class prototype augmentation (PA) to create virtual classes by combining current prototypes.

In the GCL setting, where novel classes have both few-shot and zero-shot representations, we compare against zero-shot learning frameworks with inductive learning as baselines. These approaches include:

### 9.2.3 Zero-shot learning

**DCDFL** (741, 2024): In DCDFL, a model for zero-shot node classification captures dependencies and learns discriminative features. It uses a relation-aware network to leverage long-range dependencies between nodes and employs a domain-invariant adversarial loss to reduce domain bias and promote domain-insensitive feature representations. Additionally, it enhances the representation by utilizing inter-class separability within the metric space.

**GraphCEN** (Ju et al., 2023): GraphCEN constructs an affinity graph to model class relations and uses node- and class-level contrastive learning (CL) to jointly learn node embeddings and class assignments. The two levels of CL are optimized to enhance each other.

**(CDVSc, BMVSc, WDVSc)** (Wan et al., 2019a): Based on the observation that visual features of test instances form distinct clusters, a new visual structure constraint on class centers for transductive ZSL is proposed to improve the generality of the projection function and alleviate domain shift issues. Three strategies—symmetric Chamfer distance, bipartite matching distance, and Wasserstein distance—are used to align the projected unseen semantic centers with the visual cluster centers of test instances.

**Random guess**: Randomly guessing an unseen label, introduced as a naive baseline.

### 9.3 Parameter settings

In our proposed framework, various sets of hyper-parameters are involved. These are summarized in Table 6 below. The code implementation is available here: `https://encr.pw/uYOe2`

Table 6: Parameter settings

| Parameter | Value | Parameter | Value |
|---|---|---|---|
| word2vec | 512 | $\{\alpha_1, \alpha_2, \alpha_3\}$ | $\{1, 0.25, 1\}$ |
| meta_lr | $\{1e^{-3}, 1e^{-5}\}$ | Hidden layer (MLP) | 512 |
| Random walk | $\{2, 3, 4\}$ | Hidden channels (GNNs) | 512 |
| ft_lr | $\{1e^{-3}, 1e^{-5}\}$ | Out channels (MLP) | 512 |
| Boundary $(\gamma)$ | 0.01 | Out channels (GNNs) | 512 |
| weight decay | $5e^{-3}$ | $\{\lambda_1, \lambda_2\}$ | $\{1, 1\}$ |

### 9.4 Proof of Prototype Distortion in Evolving Graphs

**Proof.** The prototype $P_i$ for a given class at any time $t$ is defined by Equation (1) in the manuscript. It represents an aggregation of embeddings from all nodes in its extended neighborhood. For simplicity, we refer to the prototype $P_i$ as the super-node $i$, and its corresponding extended support set as its neighborhood $\mathcal{N}_t(i)$ at time $t$. The embeddings are obtained as the output of the GNN model. Equivalently, the super-node $i$ can be expressed as $f_t(i; \theta)$, which represents its embedding vector at any time $t \geq 0$.

To maintain a consistent notation throughout the manuscript, we reframe the previously defined GNN embeddings for a node $v_i$, denoted as $\text{GNN}_\theta(v_i)$, as $f_t(i; \theta)$ when referring to the embedding of the super-node $i$. Therefore we have,

$$f_t(i; \theta) = \sum_{j=1}^{N} a_j \sigma \left( \frac{1}{d_t(i)} \sum_{k \in \mathcal{N}_t(i)} x_k^\top W_j + b_j \right) \tag{8}$$

Thus, the expected loss of the parameter $\theta^*$ on the node $i$ at time $t$ is

$$\Delta(P_i, \delta t) = \mathbb{E}\left[ (f_{t+1}(i; \theta) - f_t(i; \theta))^2 \right] \tag{9}$$

$$= \mathbb{E}\left[ \left( \sum_{j=1}^{N} a_j \left( \sigma\left( \frac{1}{d_{t+1}(i)} \sum_{k \in \mathcal{N}_{t+1}(i)} x_k^\top W_j + b_j \right) - \sigma\left( \frac{1}{d_t(i)} \sum_{k \in \mathcal{N}_t(i)} x_k^\top W_j + b_j \right) \right) \right)^2 \right] \tag{10}$$

Furthermore, recall that each parameter $a_j \sim U(a_j^*, \xi)$ and each element $W_{j,k}$ in the weight vector $W_j$ also satisfies $W_{j,k} \sim U(W_{j,k}^*, \xi)$. Therefore, the differences $a_j - a_j^*$ and $W_{j,k} - W_{j,k}^*$ are all i.i.d. random variables drawn from distribution $U(0, \xi)$. Therefore, we have

$$\Delta(P_i, \delta t) = \mathbb{E}\left[ \left( \sum_{j=1}^{N} a_j \left( \sigma\left( \frac{1}{d_{t+1}(i)} \sum_{k \in \mathcal{N}_{t+1}(i)} x_k^\top W_j + b_j \right) - \sigma\left( \frac{1}{d_t(i)} \sum_{k \in \mathcal{N}_t(i)} x_k^\top W_j + b_j \right) \right) \right)^2 \right] \tag{11}$$

$$= \mathbb{E}\left[ \left( \sum_{j=1}^{N} (a_j - a_j^*) \left( \sigma\left( \frac{1}{d_{t+1}(i)} \sum_{k \in \mathcal{N}_{t+1}(i)} x_k^\top W_j + b_j \right) - \sigma\left( \frac{1}{d_t(i)} \sum_{k \in \mathcal{N}_t(i)} x_k^\top W_j + b_j \right) \right) \right. \right. \tag{12}$$

$$\left. \left. + \sum_{j=1}^{N} a_j^* \left( \sigma\left( \frac{1}{d_{t+1}(i)} \sum_{k \in \mathcal{N}_{t+1}(i)} x_k^\top W_j + b_j \right) - \sigma\left( \frac{1}{d_t(i)} \sum_{k \in \mathcal{N}_t(i)} x_k^\top W_j + b_j \right) \right) \right)^2 \right] \tag{13}$$

$$= \mathbb{E}\left[ \left( \sum_{j=1}^{N} (a_j - a_j^*) \left( \sigma\left( \frac{1}{d_{t+1}(i)} \sum_{k \in \mathcal{N}_{t+1}(i)} x_k^\top W_j + b_j \right) - \sigma\left( \frac{1}{d_t(i)} \sum_{k \in \mathcal{N}_t(i)} x_k^\top W_j + b_j \right) \right) \right)^2 \right] \tag{14}$$

$$+ \mathbb{E}\left[ \left( \sum_{j=1}^{N} a_j^* \left( \sigma\left( \frac{1}{d_{t+1}(i)} \sum_{k \in \mathcal{N}_{t+1}(i)} x_k^\top W_j + b_j \right) - \sigma\left( \frac{1}{d_t(i)} \sum_{k \in \mathcal{N}_t(i)} x_k^\top W_j + b_j \right) \right) \right)^2 \right] \tag{15}$$

The third equality holds by the fact that the differences $(a_j - a_j^*)$'s are all i.i.d. random variables drawn from the uniform distribution $U(0, \xi)$. Therefore, we have

$$\Delta(P_i, \delta t) \geq \mathbb{E}\left[ \left| \sum_{j=1}^{N} (a_j - a_j^*) \left( \sigma\left( \frac{1}{d_{t+1}(i)} \sum_{k \in \mathcal{N}_{t+1}(i)} x_k^\top W_j + b_j \right) - \sigma\left( \frac{1}{d_t(i)} \sum_{k \in \mathcal{N}_t(i)} x_k^\top W_j + b_j \right) \right) \right|^2 \right] \tag{16}$$

Furthermore, since the differences $(a_j - a_j^*)$ are i.i.d. random variables drawn from the distribution $U(0, \xi)$, we must further have

$$\Delta(P_i, \delta t) \geq \mathbb{E}\left[\left|\sum_{j=1}^{N}(a_j - a_j^*)\left(\sigma\left(\frac{1}{d_{t+1}(i)}\sum_{k \in \mathcal{N}_{t+1}(i)} x_k^\top W_j + b_j\right) - \sigma\left(\frac{1}{d_t(i)}\sum_{k \in \mathcal{N}_t(i)} x_k^\top W_j + b_j\right)\right)\right|^2\right] \quad (17)$$

$$= \mathbb{E}\left[\sum_{j=1}^{N}\mathbb{E}\left[(a_j - a_j^*)^2|A_{t+1}, X_{t+1}\right]\left|\sigma\left(\frac{1}{d_{t+1}(i)}\sum_{k \in \mathcal{N}_{t+1}(i)} x_k^\top W_j + b_j\right) - \sigma\left(\frac{1}{d_t(i)}\sum_{k \in \mathcal{N}_t(i)} x_k^\top W_j + b_j\right)\right|^2\right] \quad (18)$$

$$= \frac{\xi^2}{3}\mathbb{E}\left[\sum_{j=1}^{N}\left|\sigma\left(\frac{1}{d_{t+1}(i)}\sum_{k \in \mathcal{N}_{t+1}(i)} x_k^\top W_j + b_j\right) - \sigma\left(\frac{1}{d_t(i)}\sum_{k \in \mathcal{N}_t(i)} x_k^\top W_j + b_j\right)\right|^2\right] \quad (19)$$

Furthermore, the leaky ReLU satisfies that $|\sigma(u) - \sigma(v)| \geq \beta|u - v|$. The above inequality further implies

$$\Delta(P_i, \delta t) \geq \frac{\xi^2}{3}\mathbb{E}\left[\sum_{j=1}^{N}\left(\sigma\left(\frac{1}{d_{t+1}(i)}\sum_{k \in \mathcal{N}_{t+1}(i)} x_k^\top W_j + b_j\right) - \sigma\left(\frac{1}{d_t(i)}\sum_{k \in \mathcal{N}_t(i)} x_k^\top W_j + b_j\right)\right)^2\right] \quad (20)$$

$$\geq \frac{\beta^2\xi^2}{3}\mathbb{E}\left[\sum_{j=1}^{N}\left(\frac{1}{d_{t+1}(i)}\sum_{k \in \mathcal{N}_{t+1}(i)} x_k^\top W_j + b_j - \frac{1}{d_t(i)}\sum_{k \in \mathcal{N}_t(i)} x_k^\top W_j - b_j\right)^2\right] \quad (21)$$

$$\geq \frac{\beta^2\xi^2}{3}\mathbb{E}\left[\sum_{j=1}^{N}\left(\frac{1}{d_{t+1}(i)}\sum_{k \in \mathcal{N}_{t+1}(i)\backslash\mathcal{N}_t(i)} x_k^\top W_j + \left(\frac{1}{d_{t+1}(i)} - \frac{1}{d_t(i)}\right)\sum_{k \in \mathcal{N}_t(i)} x_k^\top W_j\right)^2\right] \quad (22)$$

$$\geq \frac{\beta^2\xi^2}{3}\mathbb{E}\left[\sum_{j=1}^{N}\left(\left|\frac{1}{d_{t+1}(i)}\sum_{k \in \mathcal{N}_{t+1}(i)\backslash\mathcal{N}_t(i)} x_k^\top W_j\right|^2 + \left(\frac{1}{d_{t+1}(i)} - \frac{1}{d_t(i)}\right)\sum_{k \in \mathcal{N}_t(i)} x_k^\top W_j\right)^2\right] \quad (23)$$

$$\geq \frac{\beta^2\xi^2}{3}\mathbb{E}\left[\sum_{j=1}^{N}\left(\frac{1}{d_{t+1}(i)} - \frac{1}{d_t(i)}\right)^2\sum_{k \in \mathcal{N}_t(i)} x_k^\top W_j\right]^2 \quad (24)$$

Therefore, we have

$$\Delta(P_i, \delta t) \geq \frac{\beta^2\xi^2}{3}\mathbb{E}\left[\sum_{j=1}^{N}\left(\left(\frac{1}{d_{t+1}(i)} - \frac{1}{d_t(i)}\right)\sum_{k \in \mathcal{N}_t(i)} x_k^\top W_j\right)^2\right] \quad (25)$$

$$= \frac{\beta^2\xi^2}{3}\mathbb{E}\left[\sum_{j=1}^{N}\left(\left(\frac{1}{d_{t+1}(i)} - \frac{1}{d_t(i)}\right)\sum_{k \in \mathcal{N}_t(i)} x_k^\top(W_j - W_j^* + W_j^*)\right)^2\right] \quad (26)$$

$$= \frac{\beta^2 \xi^2}{3} \mathbb{E}\left[\sum_{j=1}^{N}\left(\left(\frac{1}{d_{t+1}(i)} - \frac{1}{d_t(i)}\right)\sum_{k\in\mathcal{N}_t(i)} x_k^\top(W_j - W_j^*)\right)^2\right] \qquad (27)$$

$$+ \frac{\beta^2 \xi^2}{3} \mathbb{E}\left[\sum_{j=1}^{N}\left(\left(\frac{1}{d_{t+1}(i)} - \frac{1}{d_t(i)}\right)\sum_{k\in\mathcal{N}_t(i)} x_k^\top W_j^*\right)^2\right] \qquad (28)$$

where the last equality comes from the fact that random vectors $(W_j - W_j^*)$ are i.i.d. random variables drawn from the uniform distribution $U(0, \xi)$ and are also independent of the graph evolution process. Therefore, we have

$$\Delta(P_i, \delta t) \geq \frac{N\beta^2\xi^2}{3} \mathbb{E}\left[\left(\left(\frac{1}{d_{t+1}(i)} - \frac{1}{d_t(i)}\right)\sum_{k\in\mathcal{N}_t(i)} x_k^\top(W_j - W_j^*)\right)^2\right] \qquad (29)$$

$$= \frac{N\beta^2\xi^4}{9} \mathbb{E}\left[\left(\frac{1}{d_{t+1}(i)} - \frac{1}{d_t(i)}\right)^2 \sum_{k\in\mathcal{N}_t(i)} \|x_k\|_2^2\right] \qquad (30)$$

Neglecting the specific activation functions and model details, we can summarize our result as:

$$\Delta(P_i, \delta t) \geq \mathbb{E}\left[\left(\frac{1}{d_{t+1}(i)} - \frac{1}{d_t(i)}\right)^2 \sum_{k\in\mathcal{N}_t(i)} \|x_k\|_2^2\right]$$

This establishes the desired bound and completes our proof.

