# OpenReview forum: "GOTHAM: Graph Class Incremental Learning Framework under Weak Supervision"
_TMLR — Accepted by TMLR_

### Review · Reviewer_dzy5 · 2024-08-12

**Summary Of Contributions:**

This paper focused on the node classification problem under weak supervision and aimed to classify nodes with limited labeled instances. It proposed the GOTHAM framework to do so compared the proposed algorithm with baseline algorithms numerically.

**Audience:**

Yes

**Claims And Evidence:**

Yes

**Requested Changes:**

I'd appreciate the author(s) to clarify the questions listed in the *Strengths And Weaknesses* section.

**Strengths And Weaknesses:**

Strengths:
1. This paper provided a good review of related works.
1. Section 3.1 formulated the problem properly.


Weaknesses:
1. May the author(s) emphasize the key difference between the proposed algorithms and existing algorithms?
1. What is the possible reason that the GOTHAM algorithm outperforms the exisiting algorithms?

---

> ### Author Response · Authors · 2024-08-27
>
> ### **Q1: May the author(s) emphasize the key difference between the proposed algorithms and existing algorithms?**
>
> ### **R1:**
> ### In this paper, we have proposed three variants of the algorithm: **Graph Orientation Through Heuristics and Meta-learning (GOTHAM)**. GOTHAM I.o and II.o have been compared against few-shot learning algorithms, while GOTHAM III.o has been compared with generalized zero-shot learning frameworks.
>
> #### **Key differences**:
>
> 1. **Prototype representation**: We represent each class with a prototype, a representative vector in the embedding space.
>
>    - **Previous approaches:** These methods typically involve directly averaging the available $k$-shot samples, leveraging attention scores, or considering node degrees to compute a weighted average.
>
>    - **Ours (GOTHAM I.o):** In addition to the $k$-shot support nodes for each class, we apply the label smoothness principle to include the local neighborhood of these nodes. This extended support set comprises labeled support nodes and their unlabeled neighbors, gathered via random walks. The final node set for a class is represented as:
>
>      $$
>      S_{x, C} = S_{C} \cup V_{C}
>      $$
>
>
>      where $S_{C}$ contains labeled nodes and $V_{C}$ consists of sampled unlabeled nodes from the seen classes. This approach enhances prototype generation.
>
> 2. **Semantic attributes**: Each modality offers a unique perspective on class representation, contributing to a more comprehensive view of prototypes.
>
>    - **Previous approaches:** Semantic attributes have primarily been used for representing zero-shot classes. Extensive experiments have utilized "class labels" and "class descriptors" to improve embeddings, but no techniques have focused on leveraging these embeddings for few-shot representation.
>
>    - **Ours (GOTHAM II.o):** We train a Multi-layer Perceptron (MLP) to encode the semantic embeddings. Combining feature-based embeddings with these semantic attributes yields a more robust class representation. Refer to Section 3.4 for more details.
>
> 3. **Handling few-shot and zero-shot classes simultaneously**: In practice, keeping up with new classes is challenging, and obtaining extensive labeled data is even harder, as annotation is time-consuming and costly. Thus, it's essential to enable models to classify nodes from both limited labeled classes and unseen classes with no labels, a scenario known as *weakly supervised*.
>
>    - **Previous approaches:** Several works (Section 2) have addressed few-shot and zero-shot representation separately, with studies on non-Euclidean data (graphs) being particularly scarce.
>
>    - **Ours (GOTHAM III.o):** Our framework handles both few-shot and zero-shot representation during incremental streaming sessions, addressing a broader objective in representation learning.
>
> ---
>
> #### Table: Key Differentiators - Baseline vs GOTHAM
>
> | **Parameter**                       | **Baselines**                                                                 | **GOTHAM (Ours)**                                                                                                                     |
> |-------------------------------------|-------------------------------------------------------------------------------|----------------------------------------------------------------------------------------------------------------------------------------|
> | **Prototype representation**        | Direct average, attention scores, weighted average                            | Extended support set through random walk sampling; Better prototype representation: **GOTHAM I.o-III.o**                               |
> | **Semantic attributes**             | Used only for zero-shot classes                                               | Learned MLP encoding with combined feature-based and semantic embeddings, utilized for both few-shot and zero-shot representation: **GOTHAM II.o** |
> | **Handling few-shot and zero-shot** | Addressed in isolation                                                        | Handled concurrently during incremental streaming sessions: **GOTHAM III.o**                                                           |
>
> ---

---

> > ### Author Response · Authors · 2024-08-27
> >
> > **Q2: What is the possible reason that the GOTHAM algorithm outperforms the existing algorithms?**
> >
> > **R2:** We believe GOTHAM outperforms the baselines due to the following reasons:
> >
> > - **Problem setting:** Graph Few-shot Class Incremental Learning (GFSCIL)
> >
> >     For the GFSCIL case, we have employed the GOTHAM I.o and II.o algorithms. The reasons they outperform the baselines are:
> >
> >     1. **Better prototype representation:** In addition to the support set, we also gather the local neighborhood through random walks. This significantly minimizes the effect of noisy data, as the average is carried out over a larger sample space.
> >
> >     2. **Use of semantic embeddings:** Integrating semantic attributes with feature-based embeddings results in better prototype representation. As a result, greater separability is achieved during self-supervised training.
> >
> > - **Problem setting:** Graph Class Incremental Learning under Weak Supervision (GCL)
> >
> >     For the GCL setting, we have employed the GOTHAM III.o framework. The reasons it outperforms the baselines are:
> >
> >     1. **Better prototype representation:** As discussed previously, semantics are used for few-shot representation as well. This ensures better separability.
> >
> >     2. **Concurrent handling of few-shot and zero-shot classes:** This framework simultaneously handles classes corresponding to few-shot and zero-shot (no training instances) representation. No other framework has addressed this setting previously.

---

### Review · Reviewer_9zXd · 2024-09-25

**Summary Of Contributions:**

The manuscript proposes a node classification method in the setting of class incremental learning with a more  realistic assumption on class labels, where new classes are introduced with a few labels or no (zero) labels. The proposed method is implemented using prototype learning based on both attribute and semantic information in the framework of meta-learning.

**Audience:**

Yes

**Broader Impact Concerns:**

No concerns on the ethical implication of the work.

**Claims And Evidence:**

Yes

**Requested Changes:**

Please revise figures and tables.
Please respond to my comments in the above section (strengths and weaknesses).

**Strengths And Weaknesses:**

Strengths
1. The manuscript is easy to read and understand.
2. The algorithm looks easy to implement.

Weaknesses
1. The novelty of the proposed method seems limited. The proposed method seems to be a combination of established tools for the node classification task. The manuscript does not propose new computational approaches, improve existing methods, or provide new interpretations of existing methods.
2. The presentation could be improved. For example, the figures seem to need annotations. Figure 1: What are the definitions of the node colors (purple and green) and what are the question marks? Figure 2: it looks better if the definitions of the shapes (e.g. stars) are given. In Table 2, the plot on the right is not mentioned in the table caption.
3. Some points are not explicitly explained in the main text. For example, the implementation of classification using protypes is not included in the text. Classification loss is also not mentioned in the algorithm section. It is not clear how the semantic attributes are calculated for the data sets in the experimental results section. I think it is unfair if this semantic information is only available for the proposed method and not for the baselines.
4. Related to point 3 above, it is not explicitly explained which aspects of the proposed method can lead to the superior performance compared to the baselines reported in the experimental results section.

---

> ### Author Response · Authors · 2024-09-28
> **Official Comments by Authors**
>
> We thank the reviewer for taking the time to review our paper. We appreciate the strengths noted and address the weaknesses as follows:
>
> **Q1: The novelty of the proposed method seems limited. The proposed method seems to be a combination of established tools for the node classification task. The manuscript does not propose new computational approaches, improve existing methods, or provide new interpretations of existing methods.**
>
> The framework presents a novel approach distinguished by two key aspects. First, it addresses a broader objective by tackling the task of Graph Class Incremental Learning under Weak Supervision, which is novel in terms of the problem statement. Second, the framework introduces innovations across different tasks, further showcasing its novelty through the specific contributions detailed below:
>
> 1. **Problem setting: Graph Few-shot Class Incremental Learning (GFSCIL)**
>
>     For the GFSCIL case we have employed the GOTHAM I.o and II.o algorithms. The reason it outperforms the baselines is:
>     - **Better prototype representation:** In addition to utilizing the support set, we also aggregate the local neighborhood through random walks. This approach significantly mitigates the impact of noisy data by averaging over a larger sample space. In many practical scenarios, the scarcity of labeled data poses a significant challenge; however, abundant unlabeled data is often available. By employing random walks to gather nodes, we can achieve a more robust representation for each prototype.
>
>     **Comparison with previous approaches:**
>     - **HAG-Meta, GEOMETER, CPCA:** These methods depend on available labeled nodes for prototype construction or leverage attention-based criteria. The former often shows greater deviation in prototype representation due to noisy data. In the latter case, even with attention mechanisms, the low number of labeled nodes leads to substantial influence from noisy data.
>
>     In contrast, our approach enhances sample size by incorporating the unlabeled neighborhood, which equalizes the contribution of each node and results in improved prototypes. **The evidence is improved accuracy when compared to baseline methods across multiple streams. Please refer to Table 2 below.**
>
> | Stream   | Base   | $S_1$  | $S_2$  | $S_3$  | $S_4$  | $S_5$  |
> |----------|--------|--------|--------|--------|--------|--------|
> | Meta-GNN | **99.60** | 86.33  | 82.43  | _77.75_ | 70.82  | 67.94  |
> | GPN      | 93.56  | 85.23  | 74.88  | 73.40  | 66.17  | 63.36  |
> | iCaRL    | 66.20  | 47.33  | 39.13  | 35.75  | 29.84  | 29.66  |
> | HAG-Meta | 95.43  | 88.76  | 75.67  | 69.56  | 67.21  | 61.86  |
> | GEOMETER | 95.44  | _90.05_ | 77.36  | 74.27  | 73.08  | **74.36** |
> | CPCA     | 95.37  | 87.88  | _83.72_ | 77.13  | _76.37_ | 69.32  |
> | **GOTHAM I.o** | _96.61_  | **90.91** | **88.89** | **84.55** | **78.82** | _73.81_ |
> | **% gain** | **-03.00** | **00.00** | **06.17** | **08.74** | **03.20** | **00.00** |
>
> *Table 2: Performance metrics for various models in the Amazon dataset under a 1-way 5-shot GFSCIL setting.*
> **(Part 2 of this answer continued further)**

---

> ### Author Response · Authors · 2024-09-28
> **Official Comments by Authors**
>
> - **Use of semantic embeddings:** Integrating the semantic attributes with feature-based embeddings results in a better prototype representation. As a result, larger separability amongst classes is achieved during incremental streaming sessions. Our evaluation within the framework shows that incorporating semantic information provides a significant advantage over methods that rely solely on feature-based embeddings.
>
>   **Comparison with previous approaches:**
>   - **HAG-Meta, GEOMETER, CPCA:** The baselines did not consider integrating semantic attributes. To our knowledge, the use of class semantic descriptors (CSDs) has been limited to zero-shot learning. With the growing trend of combining multiple modalities for richer representations, we explore the integration of semantic attributes with features to enhance representation quality. This approach not only outperforms the baselines but also offers a fresh perspective on improving existing methods. **Please refer to Table 3 (below) for more details. Additionally, Figure 4 presents an ablation study highlighting the importance of each loss term in relation to the model's accuracy.**
>
> ---
>
> ##### Table 3: **GFSCIL with semantics:** Node classification accuracy (%) in the GFSCIL setting, leveraging semantic attributes for enhanced class representation on Cora-ML and OBGN-Arxiv datasets with **GOTHAM**.
>
> | Stream | Base | S1   | S2   | S3   | S4   | S5   |
> |--------|------|------|------|------|------|------|
> | **Cora-ML (1-way 5-shot GFSCIL setting)** | | | | | | |
> | Meta-GNN   | _100_ | 79.19 | 61.37 | 60.40 | 51.51 | 36.76 |
> | GPN        | 95.58 | **91.89** | _77.95_ | 68.57 | _70.53_ | 62.53 |
> | iCaRL      | 93.00 | 69.13 | 53.81 | 47.20 | 42.86 | 38.60 |
> | HAG-Meta   | 96.08 | 87.81 | 73.96 | 70.12 | 66.19 | 60.17 |
> | GEOMETER   | 96.46 | 89.91 | 77.58 | 70.20 | 54.50 | _62.76_ |
> | CPCA       | 97.67 | 90.68 | 77.38 | _75.38_ | 69.50 | 59.86 |
> | **GOTHAM I.o** | 100   | 90.15 | 87.83 | 83.66 | 76.56 | **75.03** |
> | **GOTHAM II.o** | **100** | _91.43_ | **88.69** | **84.00** | **76.92** | 72.40 |
> | **% gain** | **00.00** | **00.00** | **13.78** | **11.43** | **09.06** | **19.55** |
>
> ---
>
> | Stream | Base | S1   | S2   | S6   | S9   | S10  |
> |--------|------|------|------|------|------|------|
> | **OBGN-Arxiv (3-way 10-shot GFSCIL setting)** | | | | | | |
> | Meta-GNN   | 76.60  | 66.10  | 57.38  | 36.55  | _29.77_ | 28.82  |
> | GPN        | 78.38  | 68.21  | 57.88  | 35.77  | 28.78  | _30.12_ |
> | iCaRL      | 62.80  | 39.54  | 35.22  | 21.97  | 15.68  | 16.45  |
> | HAG-Meta   | 77.17  | 68.19  | 58.22  | 37.13  | 28.28  | 24.68  |
> | GEOMETER   | _80.08_ | **70.68** | **61.07** | _38.13_ | 29.65  | 26.22  |
> | CPCA       | 69.71  | 56.96  | 50.39  | 33.35  | 25.76  | 24.88  |
> | **GOTHAM I.o** | 72.33  | 59.94  | 47.84  | 30.31  | 25.12  | 25.44  |
> | **GOTHAM II.o** | **82.91** | _70.20_ | _60.26_ | **40.53** | **31.38** | **32.38** |
> | **% gain** | **03.53** | **00.00** | **00.00** | **06.29** | **05.41** | **07.50** |
>
> **(Part 3 of this answer continued further)**

---

> ### Author Response · Authors · 2024-09-28
> **Official Comments by Authors**
>
> ##### 2. **Problem Setting: Graph Class Incremental Learning under Weak Supervision (GCL)**
>
> For the GCL setting, we have employed the GOTHAM III.o framework. The reason it outperforms the baselines is as follows:
>
> 1. **Better prototype representation:** As discussed previously, semantics are used for few-shot representation as well. This ensured better separability.
> 2. **Concurrent handling of few-shot and zero-shot classes:** This framework simultaneously handles classes corresponding to few-shot and zero-shot (no training instances) representation. No other framework has handled this setting previously. **Please refer to Table 4 in the paper for more details. Additionally, Figures 4A and 5 provide a deeper understanding of the contributions of our framework.**
>
> | Stream         | Base   | S1   | S2   | S3   | S4   | S5   | S6   | S7   | S8   | S9   | S10  |
> |----------------|--------|-------|-------|-------|-------|-------|-------|-------|-------|-------|-------|
> | Random guess   | 18.10  | 14.62 | 12.50 | 09.47 | 08.64 | 08.00 | 06.43 | 05.81 | 05.59 | 04.86 | 05.00 |
> | CDVSc          | 68.35  | 50.86 | 43.02 | 37.68 | 29.28 | 26.03 | 22.12 | 19.78 | 15.33 | 11.48 | 10.73 |
> | BMVSc          | 68.38  | 51.54 | 44.28 | 36.78 | 30.30 | 27.22 | 21.98 | 20.12 | 19.78 | 10.56 | 09.34 |
> | WDVSc          | 67.22  | 50.98 | 45.02 | 35.78 | 29.54 | 26.77 | 21.67 | 18.34 | 16.88 | 11.56 | 07.86 |
> | GraphCEN       | _77.13_| _62.37_| _51.76_| _38.42_| 28.92  | 18.80 | 15.36 | 10.56 | 08.92 | 08.56 | 06.53 |
> | DCDFL          | 64.66  | 53.42 | 45.06 | 32.76 | _30.48_| _30.57_| _28.44_| _25.89_| _24.18_| _22.00_| _19.62_ |
> | **GOTHAM III.o** | **82.91** | **68.11** | **59.90** | **47.68** | **43.67** | **43.31** | **38.76** | **36.41** | **34.62** | **32.55** | **30.28** |
> | **% gain**     | **07.49** | **09.20** | **15.73** | **24.10** | **43.27** | **41.67** | **36.29** | **40.63** | **43.20** | **47.95** | **54.33** |
>
> **Table 4:** Node classification accuracy (%) on the OBGN-Arxiv dataset under the GCL setting. In each streaming session, one class is designated as zero-shot, lacking any training examples.
>
> In summary, this approach introduces novelty in two key aspects: first, by addressing a broader and more practical objective, offering key insights for improving existing methods; and second, by integrating multiple modalities to enhance representation, leading to new contributions in the field.

---

> > ### Author Response · Authors · 2024-09-28
> > **Official Comments by Authors**
> >
> > **Q2: The presentation could be improved. For example, the figures seem to need annotations.
> > Figure 1: What are the definitions of the node colors (purple and green) and what are the question marks?
> > Figure 2: it looks better if the definitions of the shapes (e.g., stars) are given. In Table 2, the plot on the right is not mentioned in the table caption.**
> >
> > ---
> >
> > ### **Figure 1: Explanation**
> >
> > - At each time step, new classes with $k$-shot representations ($k = 3$ in the plot) are introduced into the existing network, illustrating the class incremental learning setup. Each class is distinguished by a different color (e.g., green, purple).
> > - The question marks (?) next to the unlabeled nodes indicate that their labels are unknown.
> >
> > ---
> >
> > ### **Figure 2: Explanation**
> >
> > In the current plot, we have labeled each shape with specific notations:
> >
> > - The star ($\star$) represents the prototype when semantic attributes are integrated with features.
> > - The plus sign ($+$) indicates the prototype when only features are used for representation.
> > - The rhombus ($\diamond$) denotes the semantic prototype, applicable in cases where features are unavailable (zero-shot learning).
> >
> > We will ensure that these labels are clearly explained in the final version of the manuscript. Thank you for your feedback.
> >
> > ---
> >
> > ### **Table 2: Explanation**
> >
> > Table 2 caption: (as in the paper)
> >
> > **GFSCIL setting:** **(Left)** Model performance (%) on the Amazon dataset under GFSCIL setting, **(Right)** Visualization of class prototypes for the Amazon dataset across different streaming sessions.
> >
> > The plot next to Table 2 visualizes the class prototypes across different streaming sessions for the Amazon dataset. This dataset contains 10 classes, with 5 as base classes, and we used a 1-way, 5-shot setting to introduce one new class in each streaming session. The line: **(Right)** Visualization of class prototypes for the Amazon dataset across different streaming sessions refers to this plot.
> >
> > If any confusion remains or further clarification is needed, we will ensure to provide a more detailed and clearly illustrated caption in the final version of the manuscript. Thank you for your feedback.
> >
> > ---
> >
> > **Q3: Some points are not explicitly explained in the main text. For example, the implementation of classification using prototypes is not included in the text. Classification loss is also not mentioned in the algorithm section. It is not clear how the semantic attributes are calculated for the datasets in the experimental results section. I think it is unfair if this semantic information is only available for the proposed method and not for the baselines.**
> >
> > ---
> >
> > - **Classification using prototypes is not included in the text. Classification loss is also not mentioned in the algorithm section.**
> >
> > During the base training stage, the framework employs the loss function $L_{train}$, which consists of three components: intra-class clustering, inter-class separation, and semantic manipulation. This loss function integrates prototypes, class samples, and semantic attributes. During fine-tuning, the framework uses $L_{finetune}$, which additionally includes knowledge distillation. Once the model is trained with these customized loss functions, classification is performed by matching the query embedding with the prototypes using cosine similarity. Thus, the loss functions themselves serve as classification losses. **Please refer to section: 4 in the manuscript.**
> >
> > ---
> >
> > - **It is not clear how the semantic attributes are calculated for the datasets in the experimental results section. I think it is unfair if this semantic information is only available for the proposed method and not for the baselines.**
> >
> > The semantic attributes refer to the encoded representations of textual descriptions for each class. These attributes can be divided into two categories: label Class Semantic Descriptors (CSDs) and Text CSDs, which utilize detailed class descriptions. In our study, encoding was performed using the widely-used "Word2Vec" framework, with relevant details provided in the **Experiment Setting** section (Section 5) of the paper.
> >
> > The datasets utilized in this research are well-established across various baselines. Zero-shot learning frameworks fundamentally depend on the availability of semantic attributes; therefore, our choice of datasets is influenced by the requirements of these tasks. To maintain fairness, we have opted to use only label CSDs, as textual CSDs rely heavily on the quality of their descriptions.
> >
> > While we acknowledge the existence of semantic attributes, they have not been explored in the context of the tasks discussed.

---

> > > ### Author Response · Authors · 2024-09-28
> > > **Official Comments by Authors**
> > >
> > > **Q4: Related to point 3 above, it is not explicitly explained which aspects of the proposed method can lead to the superior performance compared to the baselines reported in the experimental results section.**
> > >
> > > We believe our framework (GOTHAM) outperforms the baselines due to the following reasons:
> > >
> > > ---
> > >
> > > - **Problem setting: Graph Few-shot Class Incremental Learning (GFSCIL)**
> > >
> > > For the GFSCIL case, we have employed the GOTHAM I.o and II.o algorithms. The reason it outperforms the baselines is:
> > >
> > >   1. **Better prototype representation:** Apart from the support set, we additionally gather the local neighborhood through random walks. This significantly minimizes the effect of noisy data (average is carried out over a larger sample space).
> > >
> > >   2. **Use of semantic embeddings:** Integrating the semantic attributes with feature-based embeddings results in a better prototype representation. As a result, greater separability is achieved when performing self-supervised training.
> > >
> > > ---
> > >
> > > - **Problem setting: Graph Class Incremental Learning under Weak Supervision (GCL)**
> > >
> > > For the GCL setting, we have employed the GOTHAM III.o framework. The reason it outperforms the baselines is as follows:
> > >
> > >   1. **Better prototype representation:** As discussed previously, semantics are used for few-shot representation as well. This ensures better separability.
> > >
> > >   2. **Concurrent handling of few-shot and zero-shot classes:** This framework simultaneously handles classes corresponding to few-shot and zero-shot (no training instances) representation. No other framework has handled this setting previously.

---

### Review · Reviewer_VPVs · 2025-01-11

**Summary Of Contributions:**

This paper studies the problem of graph class incremental learning under the weakly supervision setting. Specifically, the new incoming data could have few or no labels. The proposed method adopts a prototype based method, so that the data are classified by being matched to the closest prototypes, which represent the corresponding classes. In this way, whether the labels of new data is abundant does not matter.

The proposed method obtains better performance than the baselines across three different datasets.

**Audience:**

Yes

**Claims And Evidence:**

Yes

**Requested Changes:**

Please refer to the weakness above.

**Strengths And Weaknesses:**

Strengths:

1. The investigated label-scarce scenario is practically important, because in practice the new incoming classes may emerge with only a few examples.

2. The proposed method can very well handle this scenario and obtains good performance.

Weakness

1. It seems that the literature review is not thorough enough. In my understanding, the class-incremental learning is also a kind of lifelong graph learning or graph continual learning. I would recommend the authors to discuss the setting of tis work with exiting graph continual learning settings, e.g. the ones in 'Cglb: Benchmark tasks for continual graph learning'. Second, the evolving nature of the graph studied in this work is also related to works like 'Temporal Generalization Estimation in Evolving Graphs'. Besides, the prototype based model also looks relevant to some other works, e.g. 'Hierarchical prototype networks for continual graph representation learning','Geometer: Graph Few-Shot Class-Incremental Learning via Prototype Representation', etc.

2. It is mentioned in the abstract that the proposed method also considers the text rich graph. Do the adopted datasets cover this scenario?

---

> ### Author Response · Authors · 2025-01-21
> **Official Comments by Authors**
>
> **Q1: It seems that the review of the literature is not thorough enough. In my understanding, class-incremental learning is also a kind of lifelong graph learning or graph continual learning. I would recommend the authors to discuss the setting of this work with existing graph continual learning settings, e.g., the ones in *CGLB: Benchmark Tasks for Continual Graph Learning*. Second, the evolving nature of the graph studied in this work is also related to works like *Temporal Generalization Estimation in Evolving Graphs*. Besides, the prototype-based model also looks relevant to some other works, e.g., *Hierarchical Prototype Networks for Continual Graph Representation Learning*, *Geometer: Graph Few-Shot Class-Incremental Learning via Prototype Representation*, etc.**
>
>
> We sincerely thank the reviewer for their time and thoughtful feedback. We acknowledge the missed citations as pointed out and have incorporated them into the revised manuscript. Moreover, we have carefully reviewed the suggested readings, and we find that they align well with the direction of our work. We have addressed these readings in the context of the queries raised, as outlined below:
>
> (**Note**: Citations are provided below in the references)
>
>
> **Part 1: Discussing the Positioning of Our Work with Regard to the Paper (1)**
>
>  **Continual Graph Learning Benchmark (CGLB):**
>
> The Continual Graph Learning Benchmark (CGLB) categorizes tasks in evolving graph structures into Continual Graph Learning (CGL), Dynamic Graph Learning (DGL), and Few-Shot Graph Learning (FSGL). CGL focuses on mitigating catastrophic forgetting without relying on past data, DGL captures temporal dynamics with access to past data, and FSGL enables rapid adaptation to new tasks using meta-learning.
>
> Tasks are further classified into graph-level vs. node-level settings and task incremental vs. class incremental learning. Class incremental learning addresses the challenge of learning new classes while retaining inter-class edges.
>
> Our work lies at the intersection of CGL and FSGL in the node-level setting, where we frame each task as a few-shot or zero-shot problem, enabling models to adapt incrementally with limited labels while preserving inter-class edges. This approach aligns with CGLB’s insights and tackles real-world challenges in evolving graph settings. We also investigate related works in few-shot, zero-shot, and incremental learning for graph-based tasks, positioning our contributions within this broader context.
>
>
>
> **Part 2: Establishing a Theoretical Understanding Presented in the Paper (2) for Our Work**
> The manuscript adopts a theoretical framework as cited in [2], for understanding the evolution and stability of prototypes in a graph neural network (GNN) model.
>
> **We have provided the entire theorem in the updated version manuscript after the conclusion section. We kindly request the reviewer’s guidance on whether this should be included in the final version of the manuscript.**
>
>
>
> **Part 3: Comparison with HPN and Geometer**
>
> For the prototype-based methods discussed, namely (1) **Hierarchical Prototype Networks for Continual Graph Representation Learning** (HPNs) and (2) **Geometer: Graph Few-Shot Class Incremental Learning via Prototype Representation**, the results for the Geometer method are detailed in Tables 2 and 3 of the manuscript. Our method consistently outperforms Geometer across all three datasets mentioned. Regarding the HPNs method, we attempted to implement the code, but encountered issues with several deprecated dependencies, preventing us from reproducing the results. We kindly request your understanding regarding this matter.

---

> ### Author Response · Authors · 2025-01-21
> **Official Comments by Authors**
>
> **Q2. In the abstract, it is mentioned that the proposed method also considers the text-rich graph. Do the adopted datasets cover this scenario?**
>
> **Yes**, the datasets cover the scenario. The datasets used in this study include **Cora-ML**, **Amazon**, and **OGBN-Arxiv**. These datasets contain **text attributes** in addition to the feature matrix \(X\) and the adjacency matrix \(A\). Furthermore, **Table 3** compares the performance of the models **GOTHAM I.0** and **GOTHAM II.0**, where GOTHAM II.0 integrates text attributes for enhanced results.
>
> **To enhance understanding, we have also included examples of datasets along with their associated textual attributes in the updated version of the manuscript. For your reference, we are also providing them here.**
>
> | **Dataset**     | **Nodes** | **Features** | **Classes** | **Class Labels**                                                                 | **Tasks**       |
> |-----------------|-----------|--------------|-------------|----------------------------------------------------------------------------------|-----------------|
> | Cora-ML         | 2,708     | 1,433        | 7           | Neural Network, Rule Learning, Reinforcement Learning, Probabilistic Methods, Theory, Genetic Algorithms, Case-based | GFSCIL, GCL     |
> | Amazon          | 13,752    | 767          | 10          | Label names Unavailable                                                        | GFSCIL          |
> | OGBN-Arxiv      | 169,343   | 128          | 40          | Arxiv cs na, Arxiv cs mm, Arxiv cs lo, Arxiv cs cy, Arxiv cs cr, Arxiv cs dc, Arxiv cs hc, Arxiv cs cv, Arxiv cs ai, ... | GFSCIL, GCL     |
>
> *Table 1: Statistics of datasets used in the experiments*
>
>
> **References:**
>
> **[1]: CGLB: Benchmark Tasks for Continual Graph Learning**
>
> **[2]: Temporal Generalization Estimation in Evolving Graphs**
>
> **[3]: Hierarchical Prototype Networks for Continual Graph Representation Learning**
>
> **[4]: Geometer: Graph Few-Shot Class-Incremental Learning via Prototype Representation**

---

> > ### Comment · Reviewer_VPVs · 2025-01-23
> >
> > Thanks for the responses from the authors. I understand the issue regarding implementing some baseline code, especially given the short response time. I don't have further concerns.

---

> > > ### Author Response · Authors · 2025-01-23
> > > **Official Comment by Authors**
> > >
> > > Thank you for your understanding. We hope the paper can be accepted in its current form. Kindly let us know if you have any other questions/suggestions

---

### Author Response · Authors · 2025-01-21
**Rebuttal Summary**

We have addressed all the queries raised by the reviewers individually. Below is a summarized version of our responses, and we hope all issues have been resolved.

**Reviewer 1 (dzy5)**

**Q1**: Key differences highlighted (GOTHAM Framework): Prototype representation, integration of semantic attributes, and simultaneous handling of few-shot and zero-shot classes.

**Q2**: Highlighted the possible reasons why GOTHAM outperforms baselines across different task settings.

**Reviewer 2 (9zxd)**

**Q1**: Highlighted the novel aspects of the proposed algorithm for both GFSCIL and GCL tasks.

**Q2**: The presentation has been updated as per suggestions: Figure 2, Table 2 (figure) are revised, and the meaning of the question mark in Figure 1 is clarified in the updated manuscript.

**Q3**: All raised questions have been addressed, with references to the relevant sections in the manuscript. The calculation of semantic attributes is also included.

**Q4**: Highlighted the possible reasons why GOTHAM outperforms baselines across different task settings.

**Reviewer 3 (VpVs)**

**Q1**: The response is provided in three parts: first, we discuss the positioning of the work in relation to [1]; second, we present the adoption of lower bound estimation based on [2] (The adopted formulation is provided in the updated manuscript, following the conclusion section.)
; and finally, we compare our approach with [3] and [4].

**Q2**: The adopted datasets address the scenario of text-rich graphs.

---

### Author Response · Authors · 2025-02-15
**Final Decision on Submission**

Dear Action Editor,

We have addressed all the reviewer's comments and provided a rebuttal summary. The manuscript has been revised accordingly, and we hope it is now suitable for acceptance in its current form.

---

### Author Response · Authors · 2025-02-28
**Final Decision regarding the Manuscript**

Dear Action Editor,
We have submitted our rebuttal addressing Reviewer 3's comments and updated the manuscript accordingly. Additionally, we have provided a rebuttal summary outlining our responses to all reviewers. The final decision was expected by February 8, 2025, but we have not yet received an update. We sincerely hope that, after the revisions made so far, the manuscript is now in its final form for acceptance. Please let us know if any further information or modifications are required from our side.

---

### Author Response · Authors · 2025-04-07
**Code Implementation**

We sincerely thank the reviewers and the Action Editor for their time and valuable feedback. We truly appreciate their insights and suggestions.

Please note that the previous code link is no longer active; the codebase has been moved to the following location:
https://github.com/adityashahane10/GOTHAM--Graph-based-Class-Incremental-Learning-Framework-under-Weak-Supervision

---

### Decision · Action_Editor_TPB2 · 2025-03-08

**Recommendation:** Accept with minor revision

**Comment:**

The recommendation is to accept the paper subject to minor revisions.  The theoretical analysis in Section 7 should not appear after the conclusion.  It should appear between Sections 4 and 5.  The proof of the lower bound in Equation 8 is missing.  Please add the proof in the appendix.  Finally, a discussion of the implication of this lower bound is needed.  The manuscript acknowledges that the degree of inaccuracy of prototype estimation will increase over time, preventing the approach to generalize properly.   Is there any future work that could be recommended to address this limitation or is this a fundamental limitation that would apply to any approach?

**Audience:**

This work will be of interest to practitioners that use graph neural networks for classification in evolving scenarios.

**Claims And Evidence:**

The paper proposes a framework for graph class incremental learning under weak supervision.  Novelty is limited in the sense that the framework combines several existing ideas, but their combination is shown to be effective empirically.  The empirical evaluation provides convincing evidence of the effectiveness of the proposed framework.  This represents a useful contribution for practitioners interested in using graph neural networks for classification in evolving scenarios.